# Learning from Heterophilic Graphs: A Spectral Theory Perspective on the Impact of Self-Loops and Parallel Edges

## Abstract

Graph heterophily poses a formidable challenge to the performance of Message-passing Graph Neural Networks (MP-GNNs). The familiar low-pass filters like Graph Convolutional Networks (GCNs) face performance degradation, which can be attributed to the blending of the messages from dissimilar neighboring nodes. The performance of the low-pass filters on heterophilic graphs still requires an in-depth analysis. In this context, we update the heterophilic graphs by adding a number of self-loops and parallel edges. We observe that eigenvalues of the graph Laplacian decrease and increase respectively by increasing the number of self-loops and parallel edges. We conduct several studies regarding the performance of GCN on various benchmark heterophilic networks by adding either self-loops or parallel edges. The studies reveal that the GCN exhibited either increasing or decreasing performance trends on adding self-loops and parallel edges. In light of the studies, we established connections between the graph spectra and the performance trends of the low-pass filters on the heterophilic graphs. The graph spectra characterize the essential intrinsic properties of the input graph like the presence of connected components, sparsity, average degree, cluster structures, etc. Our work is adept at seamlessly evaluating graph spectrum and properties by observing the performance trends of the low-pass filters without pursuing the costly eigenvalue decomposition. The theoretical foundations are also discussed to validate the impact of adding self-loops and parallel edges on the graph spectrum.

## 1 Introduction

Graph Neural Networks (Scarselli et al., 2008) made remarkable strides by achieving impeccable performance in graph-structured data. The key reason behind the immense performance superiority is the message passing (MP) framework, which enables the exchange of messages between the adjacent nodes. Before judging the narrative of the success story of the MP framework, let us first mention that graphs can be broadly categorized into two classes such as (1) *homophilic* graphs where adjacent nodes share identical class labels, and (2) *heterophilic* graphs where adjacent node labels are different from each other. The prowess of MP is mostly observed in homophilic graphs because of the tendency to blend messages from similar types of neighbors. In contrast, several studies (Zhu et al., 2020), (Zhu et al., 2021), (He et al., 2022), (Suresh et al., 2021), (Wang et al., 2022) suggest that the MP framework shows exacerbating performances on heterophilic graphs due to the influence of dissimilar messages received from neighbors.

A well-known study (Nt & Maehara, 2019) reveals that all MP-GNNs such as GCN (Kipf & Welling, 2016), GraphSage (Hamilton et al., 2017), GAT (Veličković et al., 2017), SGC (Wu et al., 2019), etc, which smooth the features of adjacent nodes, are low-pass filters. The low-pass filters successfully convert the features of the connected nodes into more similar ones compared to the features of other non-adjacent nodes. This narrates the key reason behind the successful application of low-pass filters on homophilic graphs. Applying low-pass filters on the heterophilic graphs often leads to a degradation in performance as a result of smoothing the features of the dissimilar adjacent nodes. Therefore, analyzing the performance of low-pass filters on heterophilic graphs requires more in-depth scrutiny. The low-pass filters are designed to amplify the coefficients of lower frequencies in the graph spectrum, representing the eigenvalues of the symmetrically normalized graph Laplacian. In homophilic graphs, the smoothing of node features occurs

due to the amplification of the lower frequencies of the graph spectrum. In the case of heterophilic graphs, high-pass filters sharpen the node features by amplifying the higher frequencies of the graph spectrum.

The graph spectrum entails significant information regarding structural patterns such as connected components, community structures, isolated nodes, sparsity, etc. For instance, if the set of eigenvalues contains sufficient zeros, then the network will contain more connected components or isolated nodes. The network will contain weakly (strongly) connected components if the spectrum has a higher number of low frequencies (high frequencies). Therefore, the dissection of the spectrum yields profound information relating to the spatial properties of the graphs. In this work, we investigate the dependency of the graph structure on the performance of the low-pass filters applied to heterophilic networks. We also attempt to uncover the structural properties of the existing heterophilic graphs from their spectrum. The graph spectrum is typically obtained with expensive eigenvalue decomposition which imposes unnecessary computational burden or often can be infeasible to some real-world scenarios. Therefore, we seek to devise an efficient avenue that significantly addresses the computational overhead of evaluating the graph spectrum.

Table 1: Four possible categories A, B, C, and D are presented. Each category depends on the performance trends of a low-pass filter when either self-loops or parallel edges are added to the heterophilic graph.

| | Rewiring | | |
| --- | --- | --- | --- |
| | Self-loop | Parallel edge | Category |
| Performance of LPF | Increasing (↑) | Increasing (↑) | A |
| | Increasing (↑) | Decreasing (↓) | B |
| | Decreasing (↓) | Increasing (↑) | C |
| | Decreasing (↓) | Decreasing (↓) | D |

We aim to bridge the gap by offering two simple strategies that update the graph topology by incorporating self-loops and parallel edges. After the alteration of edge connections, Graph Convolution Network (GCN), a recognized and well-adopted low-pass filter, is applied to the updated graph. We observe some interesting patterns in the performance trends of GCN when the number of self-loops or parallel edges gradually increases. The performance either monotonically improves or degrades by adding either self-loops or parallel edges. Therefore, we can have four distinct combinations of performance trends. Each combination is tagged with a category name, which is mentioned in the Table 1. The category assignment is purely dependent on the combination of performance trends in both self-loops and parallel edge addition. For instance, if GCN shows an increasing trend on self-loop addition and a decreasing trend on parallel edge addition, then the underlying dataset is categorized as "B". Furthermore, for parallel edge addition, if the trend changes to increasing, then the category will change to "A". Importantly, every category delineates a particular set of characteristics regarding the spectrum of the input graph. The performance trend of GCN can be attributed to the underlying parity between the lower and higher frequencies present in the graph spectrum. Each performance trend offers a unique insight into the parity of lower and higher frequencies, which directly links with the spatial edge connectivity of the network. We also observe that the eigenvalues of the normalized graph Laplacian decrease with the addition of self-loops in the network. Conversely, the addition of parallel edges enhances the eigenvalues of the graph Laplacian. The shrinking or expansion of the graph frequencies leads to a specific performance trend, reflecting the distribution of eigenvalues (or frequencies) in the graph spectrum. In this context, we consider 17 benchmark heterophilic graphs to predict their spectrum characteristics by observing the performance trends of GCN after adding either self-loops or parallel edges. We also offer the theoretical underpinnings of the shrinking or expansion of the eigenvalues. The phenomena are also observed empirically on the random Erdős-Rény graphs.

**Contribution** Our contributions are briefly outlined as follows,

- We provide deeper analyses pertaining to the performance of GCN, a low-pass filter, applied to 17 benchmark heterophilic graphs. We modify the graph structure with the addition of self-loops and parallel edges separately. GCN is applied to the updated graphs and observes the performance

trends. We categorize the performance trends into four categories and each graph lies in one of the four categories.

- The categorization of performance trends leads to the identification of characteristics of the graph spectrum. Different performance trends underscore the various patterns of the spectrum which reveals the properties of the networks like connected components, community structure, sparsity, etc.

- We also observe that the frequencies in the graph spectrum decrease with the addition of self-loops and frequencies increase with the addition of parallel edges. We also establish a connection between the performance trends of GCN and the shrinking or expansion of the frequencies of the graph spectrum.

- We offer extensive theoretical underpinnings for the shrinking or expansion of the frequencies of the graph spectrum with the addition of self-loops and parallel edges in the network. The detailed proofs and derivations are discussed in the Appendix.

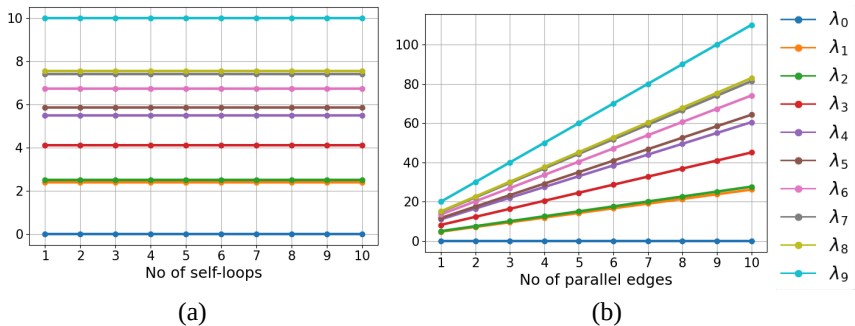

Figure 1: The changes in the eigenvalues of unnormalized graph Laplacian are presented with the addition of (a) self-loops and (b) parallel edges respectively. The eigenvalues remain unaltered with the addition of self-loops. On the contrary, the unconstrained growth of eigenvalues is observed with the addition of parallel edges.

## 2 Related Works

The spectral analysis on the graphs gains traction due to its ability to unravel the relationship between frequencies and the spatial connectivity of the networks. Work like (Ortega, 2022) introduces the key ingredients of signal processing like Graph Fourier transforms, frequencies, and the design of the filters for the graph-structured data. Another line of work (Ortega et al., 2018) deals with the intricate details of graph signal processing by shedding light on the spectrum analysis from the perspective of the graph Laplacian, extensive real-world applications, and the underlying challenges. The exploitation of spectral analysis in the discrete domain is rigorously harnessed by (Sandryhaila & Moura, 2014). Another mode of work (Tremblay et al., 2018) offers the prospect of designing versatile filter banks and spectral wavelets on graph-structured data. The design of efficient and localized convolutional filters becomes an inevitable area of research which is initiated by (Defferrard et al., 2016).

A large pool of well-adopted GNNs like GCN, GraphSage, GAT, SGC, etc are recognized as potential low-pass filters. The fact is first asserted by (Nt & Maehara, 2019). Chen *et al.* (Chen et al., 2023) established the bridge to fill the gap between the spatial and spectral properties of the prevalent graph neural networks. AutoGCN (Wu et al., 2022) proposes a variant of GCN equipped with low-pass, high-pass, and band-pass filters which automatically adjusts magnitude depending on the homophily or heterophily of the input graph. In this connection, another prominent work AdaGNN (Dong et al., 2021) learns an adaptive filter that spans across multiple layers, capturing the varying node frequencies to improve node embeddings. Taking the cue

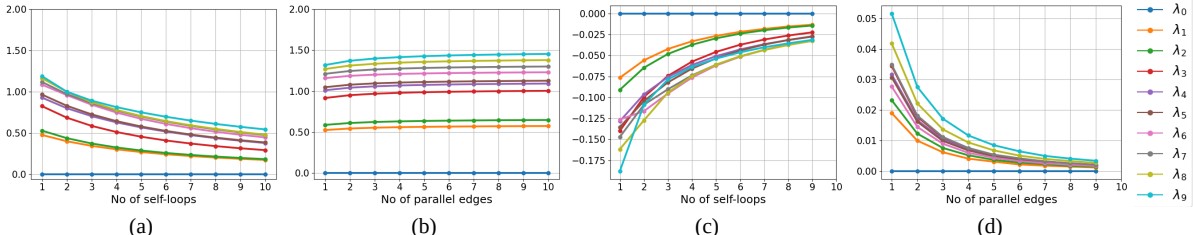

Figure 2: (a) Eigenvalues of $\tilde{L}$ exhibit a decreasing trend with the increase in addition of self-loops, and (b) eigenvalues show an increasing trend with the addition of parallel edges. The corresponding changes in the eigenvalues with the addition of self-loops and parallel edges are demonstrated in (c) and (d) respectively.

from above, FAGCN (Bo et al., 2021) firstly proposed one low-pass and one high-pass filter. The proposed filters automatically maintain the balance of collecting information from both homophilic and heterophilic graphs. Designing novel convolutional filters became an enticing avenue which is evident by (Bianchi et al., 2021) which employed an auto-regressive moving average filter (ARMA) filter by replacing polynomial filters to achieve a more robust, flexible frequency response. Another work EGC (Tailor et al., 2021) employed the effectiveness of isotropic message passing, where the message function only depends on the source nodes, over the anisotropic message passing. EGC allows the involvement of multiple learnable filters to achieve a spatially evolving frequency response. (Zhu & Koniusz, 2021) leverages the use of Markov Diffusion Kernel to obtain a filter that maintains a delicate balance between harnessing information of local and global context for each node across the network. On the other side, (Singh & Chen, 2023) identified the gap for spectral analysis on signed graphs. To mitigate the gap, they proposed two signed GNNs that retain low-pass and high-pass information respectively. Compatible Label Propagation (CLP) (Zhong et al., 2022) combines a class compatibility matrix and label propagation to improve node classification on heterophilic graphs. Very recently, an inductive spectral filter SLOG (Xu et al.) was proposed which considers real-valued order polynomial filters. SLOG also combines subgraph sampling in the spatial domain and signal processing in the spectral domain. Another orthogonal work (Guo et al., 2018) studied the effect on the eigenvalues of the adjacency matrix but no theoretical analysis was provided on the graph Laplacian or graph spectrum. Moreover, they offered analyses on the eigenvalues of the adjacency matrix for the addition of any edges. In contrast, our work predominantly presents systematic addition of self-loops and parallel edges which offers insights into the graph spectra in the context of heterophilic graphs.

## 3 A Deeper Investigation

### 3.1 Notations

Consider an attributed graph $\mathcal{G} = (\mathcal{V}, \mathcal{E}, X)$ where $\mathcal{V}$ denotes the set of vertices with $|\mathcal{V}| = n$, $\mathcal{E} \subseteq \mathcal{V} \times \mathcal{V}$ is the set of edges, and $X \in \mathbb{R}^{n \times d}$ is the feature matrix contains $d$-dimensional feature vectors. We define graph Laplacian $L = D - A$ and symmetrically normalized Laplacian as $\tilde{L} = D^{-\frac{1}{2}} L D^{-\frac{1}{2}} = \mathrm{I} - D^{-\frac{1}{2}} A D^{-\frac{1}{2}}$. Also, augmenting self-loops the normalized Laplacian will be $\tilde{L} = \mathrm{I} - \tilde{D}^{-\frac{1}{2}} \tilde{A} \tilde{D}^{-\frac{1}{2}}$ where $\tilde{A} = A + \mathrm{I}$ and $\tilde{D} = D + \mathrm{I}$. More intricate details on notations are available in Table 2.

### 3.2 Preliminaries on Spectral Graph Theory

The spectral analysis of graphs (Shuman et al., 2013) revolves around understanding the characteristics of eigenvalues and eigenvectors of the symmetrically normalized graph Laplacian $\tilde{L}$. The analysis entails that the eigenvalues and eigenvectors of $\tilde{L}$ represent the Fourier frequency and Fourier modes. Suppose the eigendecomposition on Laplacian yields us $\tilde{L} = U \Sigma U^{\top}$ where columns of $U$ represent the eigenvectors and $\Sigma$ is a diagonal matrix containing the eigenvalues. Let a signal $x \in \mathbb{R}^n$ act on the nodes in the graph, then the Fourier transformation of $x$ is presented as $\hat{x} = U^{\top} x$. The inverse Fourier transform can be formulated

as $x = U\tilde{x}$. Thus, for any filter $g$, the graph convolution between $g$ and $x$ is estimated as:

$$g * x = U((U^\top g) \odot (U^\top x)) = U\tilde{G}U^\top x, \tag{1}$$

where $\odot$ denotes the element-wise vector multiplication and $\tilde{G} = \text{diag}\{\tilde{g}_1, \cdots, \tilde{g}_n\}$. Each $\tilde{g}_i$ denotes the spectral filter coefficient. A well-known fact is that $\tilde{L}$ has the eigenvalues lie in $[0, 2]$. Let us categorize the set of eigenvalues as $\lambda_{<1}$ or *lower frequencies* which are strictly smaller than 1 and $\lambda_{\geq 1}$ or *higher frequencies*, which are greater than or equal to 1. If a filter amplifies the coefficients of $\lambda_{<1}$, then it acts as a low-pass filter, and on the contrary high-pass filter amplifies the coefficients of $\lambda_{\geq 1}$. For instance, the filter function of GCN is $\tilde{G} = \text{I} - \Sigma$ where the coefficients of $\lambda_{<1}$ increases. Therefore, GCN acts as a low-pass filter.

Table 2: The table represents all possible notations used in this work are presented with descriptions.

| Notation | Decription |
|---|---|
| A | Adjacency matrix |
| D | Degree matrix |
| $L = D - A$ | Unnomalized graph Laplacian |
| $\tilde{A} = A + I$ | Adjacency matrix augmented with self-loop |
| $A_N = D^{-\frac{1}{2}} A D^{-\frac{1}{2}}$ | Symmetrically normalized adjacency matrix |
| $\tilde{D} = D + I$ | Degree matrix augmented with self-loop |
| $\tilde{A}_\alpha = A + \alpha I$ | Adjacency matrix augmented with $\alpha$-time self-loops |
| $\tilde{D}_\alpha = D + \alpha I$ | Degree matrix augmented with $\alpha$-time self-loops |
| $\tilde{A}_\gamma = (\gamma + 1)A + \text{I}$ | Adjacency matrix augmented with $\gamma$-times parallel edges with one self-loops |
| $\tilde{D}_\gamma = (\gamma + 1)D + \text{I}$ | Degree matrix augmented with $\gamma$-times parallel edges with one self-loops |
| $\tilde{A}_N^\alpha = \tilde{D}_\alpha^{-\frac{1}{2}} \tilde{A}_\alpha \tilde{D}_\alpha^{-\frac{1}{2}}$ | Symmetrically normalized adjacency matrix augmented with $\alpha$-times self-loops |
| $\tilde{A}_N^\gamma = \tilde{D}_\gamma^{-\frac{1}{2}} \tilde{A}_\gamma \tilde{D}_\gamma^{-\frac{1}{2}}$ | Symmetrically normalized adjacency matrix augmented with $\gamma$-times parallel edges |
| $\tilde{L}_\alpha = \text{I} - \tilde{D}_\alpha^{-\frac{1}{2}} \tilde{A}_\alpha \tilde{D}_\alpha^{-\frac{1}{2}}$ | Symmetrically normalized graph Laplacian augmented with $\alpha$-times self-loops |
| $\tilde{L}_\gamma = \text{I} - \tilde{D}_\gamma^{-\frac{1}{2}} \tilde{A}_\gamma \tilde{D}_\gamma^{-\frac{1}{2}}$ | Symmetrically normalized graph Laplacian augmented with $\gamma$-times parallel edges |

### 3.3 Addition of Self-loops

We conduct experiments by adding the self-loops corresponding to each node in the graph. The number of self-loops is denoted by $\alpha$. After the addition of self-loops $\alpha$-times, the adjacency matrix will be $\tilde{A}_\alpha = A + \alpha \text{I}$ and the corresponding degree matrix will look like $\tilde{D}_\alpha = D + \alpha \text{I}$. Therefore, the symmetrically normalized graph Laplacian can be presented as $\tilde{L}_\alpha = \text{I} - \tilde{D}_\alpha^{-\frac{1}{2}} \tilde{A}_\alpha \tilde{D}_\alpha^{-\frac{1}{2}}$.

### 3.4 Addition of Parallel edges

We also perform experiments by adding parallel edges corresponding to every edge in the graph. Assume $\gamma$ denotes the number of parallel edges to be added in the network. Therefore, adding $\gamma$-times parallel edges, the updated adjacency matrix will be $A_\gamma = (\gamma + 1)A$. Thus, the corresponding degree matrix will be $D_\gamma = (\gamma + 1)D$. Adding self-loops the further modified adjacency and degree matrices will be $\tilde{A}_\gamma = (\gamma + 1)A + \text{I}$ and $\tilde{D}_\gamma = (\gamma + 1)D + \text{I}$. Therefore, we can define the corresponding symmetrically normalized graph Laplacian as $\tilde{L}_\gamma = \text{I} - \tilde{D}_\gamma^{-\frac{1}{2}} \tilde{A}_\gamma \tilde{D}_\gamma^{-\frac{1}{2}}$.

### 3.5 Empirical Evidence on Random Graphs

We conducted experiments on randomly generated Erdős-Rényi graphs to study the effects on the eigenvalues with the addition of self-loops and parallel edges in the graph. A random graph $\mathcal{G}_{er}$ is generated with 10 vertices and having edge probability 0.50. Two individual experiments are performed with (1) the addition of self-loops, and (2) the addition of parallel edges. We varied the number of self-loops or parallel edges ranging from 1 to 10. The eigendecomposition is performed for every stage of addition to monitor the changes that

Table 3: Total number of nodes, isolated nodes, edge density ($\mathrm{sp}_\mathcal{G}$), and average degree ($d_\mathrm{avg}$) for 17 heterophilic graphs are presented. We also mentioned the log-scaled versions of edge density ($\overline{\mathrm{sp}}_\mathcal{G}$) and average degrees ($\overline{d}_\mathrm{avg}$) with lower values indicate higher density and average degree.

| Properties/Datasets | Cornell | Texas | Wisconsin | Chameleon | Squirrel | Actor |
|---|---|---|---|---|---|---|
| # nodes | 183 | 183 | 251 | 2277 | 5201 | 7600 |
| # isolated nodes (%) | 87(47.5%) | 73(39.8%) | 81(32.3%) | 0(0%) | 0(0%) | 636(8.36%) |
| density ($\mathrm{sp}_\mathcal{G}/\overline{\mathrm{sp}}_\mathcal{G}$) | 0.017/4.02 | 0.019/3.93 | 0.016/4.10 | 0.013/4.27 | 0.016/4.13 | 0.001/6.86 |
| avg. degree ($d_\mathrm{avg}/\overline{d}_\mathrm{avg}$) | 1.62/3.82 | 1.77/3.82 | 2.05/4.13 | 15.85/6.34 | 41.73/7.17 | 3.94/7.54 |

| Properties/Datasets | arxiv-year | snap-patents | Penn94 | pokec | twitch-gamers | genius |
|---|---|---|---|---|---|---|
| # nodes | 169343 | 2923922 | 41554 | 1632803 | 168114 | 421961 |
| # isolated nodes (%) | 17440(10.29%) | 881754(30.15%) | 0(0%) | 200110(12.25%) | 44596(26.52%) | 371870(88.12%) |
| density ($\mathrm{sp}_\mathcal{G}/\overline{\mathrm{sp}}_\mathcal{G}$) | 8.1e-5/9.41 | 3.2e-6/12.63 | 0.003/5.75 | 3.3e-5/10.68 | 9.6e-4/7.63 | 2e-5/11.41 |
| avg. degree ($d_\mathrm{avg}/\overline{d}_\mathrm{avg}$) | 6.88/10.65 | 4.77/13.50 | 65.56/9.24 | 27.31/12.91 | 80.86/10.64 | 4.37/11.56 |

| Properties/Datasets | Roman-empire | Amazon-ratings | Minesweeper | Tolokers | Questions | - |
|---|---|---|---|---|---|---|
| # nodes | 22662 | 24492 | 10000 | 11758 | 48921 | |
| # isolated nodes (%) | 0(0%) | 3495(14.26%) | 1(0.01%) | 936(7.96%) | 17761(36.3%) | |
| density ($\mathrm{sp}_\mathcal{G}/\overline{\mathrm{sp}}_\mathcal{G}$) | 2.5e-4/8.26 | 3.1e-4/8.07 | 7.8e-4/7.14 | 0.007/4.89 | 1.2e-4/8.96 | |
| avg. degree ($d_\mathrm{avg}/\overline{d}_\mathrm{avg}$) | 2.91/8.64 | 3.79/8.71 | 3.94/7.82 | 44.14/7.98 | 3.14/9.41 | |

occurred in the corresponding eigenvalues. Refer to Figure 2 for the portrayal of the effect on the eigenvalues and the other corresponding changes in the spectrum. The plots (a) and (b) demonstrate the effect on the of 10 eigenvalues of $\mathcal{G}_{er}$ while the addition of self-loops or parallel edges in the graph. On the other side, plots (c) and (d) depict the changes in the eigenvalues. As we have added 10 self-loops or parallel edges, thus 9 differences are recorded in the plots.

**Observation** The plots reaffirm that the addition of self-loops leads to the shrinking of the eigenvalues in the spectrum except for the eigenvalue 0 (Refer Figure 2(a)). On the contrary, eigenvalues increase with the addition of parallel edges (Refer Figure 2(b)). Additionally, we also present the change in the eigenvalues in Figures 2(c) and 2(d). Notably, the monotone increase (decrease) of the curves underlines the slower rate with the addition of self-loops (parallel edges).

Table 4: GCN (a low-pass filter) is applied on the 6 standard heterophilic datasets curated by Pie *et al.* (Pei et al., 2020). Performance analyses are demonstrated after adding multiple self-loops and parallel edges respectively in the graphs. The performance trends are also highlighted and corresponding categories are marked for each dataset.

| Method | Input adjacency | Chameleon | Squirrel | Actor | Cornell | Texas | Wisconsin |
|---|---|---|---|---|---|---|---|
| GCN | A + I | $71.68 \pm 1.92$ | $62.78 \pm 1.99$ | $27.40 \pm 1.12$ | $40.27 \pm 6.44$ | $54.86 \pm 5.27$ | $45.29 \pm 6.10$ |
| | A + 2I | $67.69 \pm 2.25$ | $58.64 \pm 2.27$ | $29.20 \pm 1.13$ | $43.78 \pm 5.09$ | $56.21 \pm 4.95$ | $50.19 \pm 6.39$ |
| | A + 3I | $65.15 \pm 1.56$ | $55.35 \pm 1.76$ | $30.67 \pm 1.25$ | $47.83 \pm 8.11$ | $54.05 \pm 4.18$ | $56.47 \pm 3.80$ |
| | A + 4I | $63.22 \pm 1.22$ | $53.43 \pm 1.40$ | $32.08 \pm 1.20$ | $46.48 \pm 7.81$ | $58.64 \pm 6.16$ | $60.98 \pm 4.37$ |
| | A + 5I | $61.90 \pm 2.51$ | $51.44 \pm 1.51$ | $32.94 \pm 0.85$ | $50.27 \pm 7.47$ | $57.02 \pm 7.49$ | $61.96 \pm 5.42$ |
| Trend | $\rightarrow$ | Decreasing (↓) | Decreasing (↓) | Increasing (↑) | Increasing (↑) | Increasing (↑) | Increasing (↑) |
| GCN | A + I | $71.68 \pm 1.92$ | $62.70 \pm 1.99$ | $27.40 \pm 1.12$ | $40.27 \pm 6.44$ | $54.86 \pm 5.27$ | $45.29 \pm 6.10$ |
| | 2A + I | $75.06 \pm 1.24$ | $66.43 \pm 2.40$ | $25.54 \pm 1.30$ | $40.54 \pm 6.39$ | $54.59 \pm 6.13$ | $47.45 \pm 4.94$ |
| | 3A + I | $76.31 \pm 0.99$ | $67.79 \pm 1.96$ | $25.63 \pm 0.69$ | $44.05 \pm 7.83$ | $55.67 \pm 5.43$ | $47.25 \pm 5.22$ |
| | 4A + I | $76.60 \pm 0.77$ | $67.92 \pm 1.66$ | $24.76 \pm 1.19$ | $45.67 \pm 8.58$ | $51.89 \pm 7.43$ | $46.86 \pm 4.24$ |
| | 5A + I | $77.32 \pm 1.07$ | $68.59 \pm 1.71$ | $24.82 \pm 1.34$ | $41.08 \pm 5.64$ | $51.08 \pm 10.0$ | $46.86 \pm 4.75$ |
| Trend | $\rightarrow$ | Increasing (↑) | Increasing (↑) | Decreasing (↓) | Increasing (↑) | Increasing (↑) | Increasing (↑) |
| Category | $\rightarrow$ | C | C | B | A | A | A |

### 3.6 Theoretical Analysis: A Spectral Perspective

We perform a deeper theoretical analysis of the effect on the graph spectrum when adding self-loops or parallel edges. The detailed study is provided as follows,

**Lemma 1.** *Consider a graph $G$ with $A$ and $D$ as the adjacency and degree matrix. Now $\alpha$-times self-loops are added in $G$ with $\alpha_1 \in \mathbb{Z}^+$. Assume $\lambda_\alpha^{max}$ is the maximum eigenvalue of symmetrically normalized*

graph Laplacian $\tilde{L}_\alpha$ of the updated graph. If $\beta_1$ is the smallest eigenvalue of $D^{-\frac{1}{2}} A D^{-\frac{1}{2}}$ and $\max_i d_i$ is the maximum degree of $G$, then $\lambda_\alpha^{max} \leq \frac{\max_i d_i (1-\beta_1)}{\alpha + \max_i d_i}$.

**Lemma 2.** *Consider a graph $G$ with $A$ and $D$ as the adjacency and degree matrix. Now $\gamma$-times self-loops are added in $G$ with $\gamma \in \mathbb{Z}^+$. Assume $\lambda_\gamma^{max}$ is the maximum eigenvalue of symmetrically normalized graph Laplacian $\tilde{L}_\gamma$ of the updated graph. If $\beta_1$ is the smallest eigenvalue of $D^{-\frac{1}{2}} A D^{-\frac{1}{2}}$ and $\max_i d_i$ is the maximum degree of $G$, then $\lambda_\gamma^{max} \leq \frac{(1+\gamma) \max_i d_i (1-\beta_1)}{1 + (1+\gamma) \max_i d_i}$.*

**Remark 1.** *The maximum eigenvalue of $\tilde{L}_\alpha$ decreases with the increasing number of self-loops in the network, indicating the shrinking of the graph spectrum. On the contrary, the maximum eigenvalue of $\tilde{L}_\gamma$ increases with the increasing number of parallel edges in the network, illustrating the expansion of the graph spectrum.*

**Lemma 3.** *Given a $k$-regular graph $\mathcal{G}$, the eigenvalues of $\tilde{A}_N^\alpha$ will lie in $[-1, 1]$ $\forall \alpha \geq 1$.*

**Lemma 4.** *Given a $k$-regular graph $\mathcal{G}$, the eigenvalues of $\tilde{A}_N^\gamma$ will lie in $[-1, 1]$ $\forall \gamma \geq 1$.*

**Remark 2.** *The eigenvalues will lie in $[-1, 1]$ whether the self-loops or parallel edges are added in a regular graph. The range of eigenvalues will remain unaffected with the addition of self-loops or parallel edges.*

**Theorem 1.** *Consider a $k$-regular graph with $\alpha_1, \alpha_2 \in \mathbb{R}^+$ where $\alpha_1 \leq \alpha_2$, then the inequality will hold $\lambda_{\alpha_1}^i \geq \lambda_{\alpha_2}^i, \forall\ 1 \leq i \leq n$ where $\lambda_{\alpha_1}^i$ and $\lambda_{\alpha_2}^i$ are the $i^{th}$ eigenvalues of $\tilde{L}_{\alpha_1}$ and $\tilde{L}_{\alpha_2}$ respectively.*

**Theorem 2.** *Consider a $k$-regular graph with $\gamma_1, \gamma_2 \in \mathbb{R}^+$ with $\gamma_1 \leq \gamma_2$, then the inequality will hold $\lambda_{\gamma_1}^i \leq \lambda_{\gamma_2}^i, \forall\ 1 \leq i \leq n$ where $\lambda_{\gamma_1}^i$ and $\lambda_{\gamma_2}^i$ are the $i^{th}$ eigenvalues of $\tilde{L}_{\gamma_1}$ and $\tilde{L}_{\gamma_2}$ respectively.*

**Remark 3.** *We have shown that for the regular graphs, the eigenvalues of the symmetrically normalized graph Laplacian decrease concurrently when self-loops are added. Adding self-loops attenuates the graph spectrum frequencies, thereby shifting the graph spectrum towards zero eigenvalue. This will also increase the number of lower frequencies and decrease the number of higher frequencies.*

*The eigenvalues of the symmetrically normalized graph Laplacian increase when parallel edges are added. Adding parallel edges amplifies the frequencies of the graph spectrum, which shifts the spectrum toward the value 2. Consequently, the number of lower frequencies decreases and the number of higher frequencies increases.*

**Corollary 1.** *The increase in the eigenvalues of $L_\gamma$ is independent of the number of self-loop additions in $\mathcal{G}$. On the contrary, the eigenvalues of $\tilde{L}_\gamma$ will increase if at least one self-loop is added per node in $\mathcal{G}$.*

In the following two theorems, we will demonstrate the alteration of the eigenvalues of $A_N$ with the addition of self-loops or parallel edges by the perturbation of the adjacency matrix.

**Theorem 3.** *Consider a connected graph $\mathcal{G}$ with $A_N = D^{-\frac{1}{2}} A D^{-\frac{1}{2}}$. Assuming the diagonal of $A$ of $G$ is perturbed by a significantly small $\alpha > 0$, then the updated normalized adjacency matrix will be $A_N^\alpha$. The change in the eigenvalues of $A_N^\alpha$ with respect to the eigenvalues of $A_N$ will increase when $\alpha$ increases.*

**Theorem 4.** *Consider a connected graph $\mathcal{G}$ with normalized adjacency matrix $A_N = D^{-\frac{1}{2}} A D^{-\frac{1}{2}}$. Assuming each element of $A$ except the diagonal is multiplied by $1 + \gamma$ where $\gamma > 0$ is a significantly small quantity, then the updated normalized adjacency matrix will be $A_N^\gamma$. The change in the eigenvalues of $A_N^\gamma$ with respect to the eigenvalues of $A_N$ will decrease when $\gamma$ increases.*

**Remark 4.** *Theorem 3 suggests that the change in the eigenvalues of the normalized adjacency matrix increases an increasing number of self-loops which also signify the decrease in the change in the eigenvalues of $\tilde{L}_\alpha$. The small perturbation in the adjacency matrix leads to a shrinking of the frequencies in the graph spectrum. The spectrum will shift toward the zero eigenvalue. Spectrum shrinking also highlights the alteration in the parity of frequencies. The number of lower frequencies increases and the number of higher frequencies decreases.*

*Conversely, Theorem 4 states that the change of eigenvalues of the normalized adjacency matrix decreases with the increasing number of parallel edges. This indicates an increase in the change in the eigenvalues of $\tilde{L}_\gamma$, highlighting the spectrum expansion. To this effect, the graph spectrum will shift toward the value 2. Furthermore, the parity of frequencies changes, specifically, the number of lower frequencies decreases and the number of higher frequencies increases.*

### 3.7 Effect on the Performance of GCN for Spectral Shifts

The theoretical analyses offer insights into the effects on graph spectra when self-loops or parallel edges are added. We also observed that spectrum shifts either to zero eigenvalue or to the value 2 and also alter the parity of the frequencies or eigenvalues. The higher number of low frequencies in a network indicates the presence of well-separated communities, mostly sharing identical node labels. GCN or any low-pass filter is supposed to smooth out the features of the connected nodes, assuming that they share similar class labels. The addition of self-loops increases the number of lower frequencies which appears to be beneficial for smoothing out features, improving the performance of GCN.

Conversely, the graph spectra shift toward the value 2, signifying the increase in the higher frequencies. A network with overlapped communities contains many high frequencies in the spectrum. GCN is poised to smooth out the features of nodes that are probably having different class labels. This will lead to the degradation of the performance of GCN.

### 3.8 Connection between Theoretical Analysis and Performance Trends

The various performance trends depend on the presence of parity between lower and higher frequencies in the spectrum of the input graph. As discussed in the theoretical analyses, the addition of self-loops and parallel edges respectively decreases and increases the eigenvalues or frequencies in the spectrum. If GCN witnessed an increasing trend on a heterophilic graph with the gradual addition of self-loops, then we can conclude that initially the graph spectrum is aligned toward zero eigenvalue and the addition of self-loops further tilts the balance to lower frequencies. The similar effects are evident in datasets like Actor, Cornell, pokec, etc. In contrast, if the performance of GCN exacerbates with the gradual addition of parallel edges, then we can infer initial graph spectrum is skewed towards the maximum value 2, indicating a greater number of higher frequencies. Further addition of parallel edges enhances the high frequencies and the decreasing trend observed in the performance of GCN. The impact is evident in datasets like Actor, arxiv-year, snap-patents, Tolokers, etc.

## 4 Experiments

### 4.1 Datasets

Our experiments encompass three categories of datasets (1) Pie *et al.* (Pei et al., 2020) proposed 6 standard heterophilic networks consisting of Cornell, Texas, Wisconsin, Chameleon, Squirrel, and Actor, (2) Lim *et al.* (Lim et al., 2021) curated 6 large-scale heterophilic networks namely arxiv-year, snap-patents, Penn94, pokec, twitch-gamers, and genius, and (3) Platonov *et al.* (Platonov et al., 2023) identified prevailing shortcomings on the existing datasets and developed a set of 5 heterophilic networks viz Roman-empire, Amazon-ratings, Minesweeper, Tolokers, and Questions. The details of all datasets are vividly available in Table 3.

### 4.2 Experimental Settings

For all graphs from Pie *et al.*, we have considered 10 standard train/valid/test splits with 60%/20%/20% samples. Graphs from Lim *et al.* and Platonov *et al.* train/valid/test splits are fixed as 50%/25%/25% for 5 and 10 splits respectively. We applied a two-layered GCN architecture across all 17 graphs to carry out entire experiments. The dropout rate is fixed at 0.50 and LayerNorm is employed to make training convergence faster. The model parameters are optimized by Adam optimizer. We evaluated the best model on every split and finally reported mean and standard deviations across all splits for each of the datasets. The performance metric is test accuracy and for Minesweeper, Tolokers, and Questions the ROC-AUC is reported. Our Pytorch and Pytorch-geoemtric based implementation is available at https://anonymous.4open.science/r/DIHG-5212/README.md.

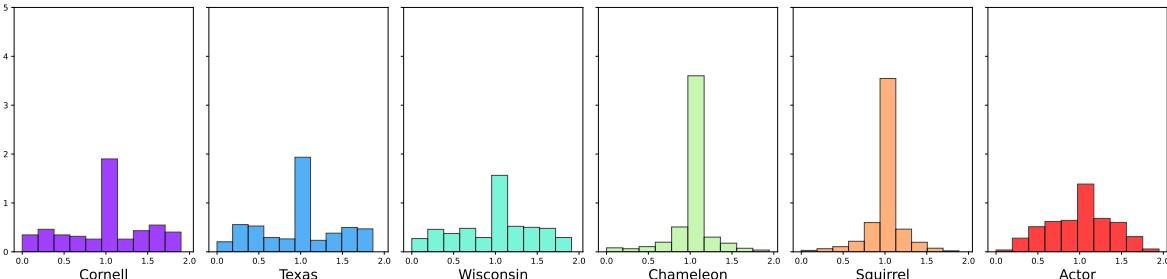

Figure 3: The distribution of eigenvalues of the normalized graph Laplacian presented for six standard heterophilic datasets obtained from (Pei et al., 2020).

### 4.3 Analysis of Isolated Nodes, Sparsity, and Average Degree in the Heterophilic Networks

We offer an in-depth analysis of the number of isolated nodes, edge density, and the average degree in the context of the heterophilic networks. Refer to Table 3 for illustrating the comparative statistics of the various networks. As per the existing formula, the edge density can be defined as $\text{sp}_{\mathcal{G}} = \frac{2|\mathcal{E}|}{|\mathcal{V}|(|\mathcal{V}|-1)}$. If the input graph is too sparse, the estimated value will be too little to comprehend. Therefore, we devise an alternative solution with the assistance of a logarithmic scale which is defined as $\overline{\text{sp}}_{\mathcal{G}} = -\log(\text{sp} + \epsilon)$ where $\epsilon$ is tiny real number is added to avert the numerical instability. The scaling suggests that the lower the $\overline{\text{sp}}_{\mathcal{G}}$, the more dense the graph is. A similar issue is confronted in the estimation of average degree as $d_{\text{avg}} = \frac{2|\mathcal{E}|}{|\mathcal{V}|}$. Therefore, we also pursue similar tricks to tackle the issue. The scaled average degree is defined as $\overline{d}_{\text{avg}} = -\log(d_{\text{avg}} + \epsilon)$ where $\epsilon$ is same as defined as earlier. A lower $\overline{d}_{\text{avg}}$ signifies the higher average degree of the underlying network.

### 4.4 Category of Distribution of Eigenvalues

Suppose we attempt to apply a low-pass filter (like GCN) on any input graph. The graph will be pre-processed either by adding a fixed number of self-loops or parallel edges. Considering all possibilities, the low-pass filter can exhibit four distinct types of performance trends. The various possible trends are vividly portrayed in Table 1. We marked the categories respectively as A, B, C, and D. The different performance trends may be the manifestations of the underlying edge connectivity of the network. The structure of the networks has an inherent connection with the eigenvalues derived from the normalized graph Laplacian. Therefore, the defined categories may help to comprehend the parity of lower and higher frequencies (eigenvalues) of the graph spectrum.

Table 5: GCN (a low-pass filter) is applied on the 6 large-scale heterophilic datasets curated by Lim *et al.* (Lim et al., 2021). Performance analyses are demonstrated after adding multiple self-loops and parallel edges respectively in the graphs. The performance trends are also highlighted and corresponding categories are marked for each dataset.

| Method | Input Adjacency | arxiv-year | snap-patents | Penn94 | pokec | twitch-gamers | genius |
|---|---|---|---|---|---|---|---|
| GCN | A + I | $48.75 \pm 0.43$ | $34.96 \pm 0.15$ | $77.12 \pm 0.43$ | $59.53 \pm 0.18$ | $60.83 \pm 0.29$ | $80.03 \pm 0.66$ |
| | A + 2I | $48.59 \pm 0.53$ | $34.77 \pm 0.14$ | $75.82 \pm 0.45$ | $59.17 \pm 0.20$ | $61.00 \pm 0.34$ | $79.81 \pm 1.38$ |
| | A + 3I | $47.34 \pm 0.20$ | $34.60 \pm 0.04$ | $74, 32 \pm 0.49$ | $59.77 \pm 0.15$ | $61.07 \pm 0.33$ | $78.77 \pm 0.99$ |
| | A + 4I | $46.26 \pm 0.21$ | $34.57 \pm 0.11$ | $72.79 \pm 0.55$ | $60.43 \pm 0.14$ | $61.18 \pm 0.26$ | $77.80 \pm 0.51$ |
| | A + 5I | $45.73 \pm 0.48$ | $34.43 \pm 0.17$ | $71.39 \pm 0.40$ | $60.94 \pm 0.17$ | $61.23 \pm 0.30$ | $77.35 \pm 0.37$ |
| Trend | $\rightarrow$ | Decreasing (↓) | Decreasing (↓) | Decreasing (↓) | Increasing (↑) | Increasing (↑) | Decreasing (↓) |
| GCN | A + I | $48.69 \pm 0.37$ | $35.00 \pm 0.15$ | $77.13 \pm 0.39$ | $59.54 \pm 0.23$ | $60.86 \pm 0.30$ | $80.09 \pm 0.70$ |
| | 2A + I | $48.31 \pm 0.54$ | $34.89 \pm 0.12$ | $77.64 \pm 0.41$ | $61.32 \pm 0.14$ | $60.72 \pm 0.21$ | $74.41 \pm 2.07$ |
| | 3A + I | $47.93 \pm 0.64$ | $34.64 \pm 0.23$ | $77.81 \pm 0.38$ | $62.22 \pm 0.10$ | $60.70 \pm 0.24$ | $69.88 \pm 0.48$ |
| | 4A + I | $47.74 \pm 0.52$ | $34.46 \pm 0.18$ | $77.88 \pm 0.37$ | $62.77 \pm 0.09$ | $60.70 \pm 0.21$ | $70.17 \pm 0.89$ |
| | 5A + I | $47.78 \pm 0.38$ | $34.28 \pm 0.13$ | $77.81 \pm 0.35$ | $63.10 \pm 0.10$ | $60.69 \pm 0.21$ | $71.13 \pm 1.26$ |
| Trend | $\rightarrow$ | Decreasing (↓) | Decreasing (↓) | Increasing (↑) | Increasing (↑) | Decreasing (↓) | Decreasing (↓) |
| Category | $\rightarrow$ | D | D | C | A | B | D |

### 4.5 Initial Distribution of Eigenvalues

We estimate the distribution of eigenvalues from the normalized graph Laplacian for six standard heterophilic datasets Cornell, Texas, Wisconsin, Chameleon, Squirrel, and Actor, Refer to Figure 3 for the detailed illustration. The histograms of Cornell, Texas, and Wisconsin are identical. On the other side, the corresponding histograms of Chameleon and Squirrel carry similar patterns, The histogram of the Actor dataset is completely different compared to the rest of the others. Later we will observe that the datasets have histograms of similar patterns that will yield identical trends in the performances.

### 4.6 Performance Categorisation of Low-pass Filter

We performed semi-supervised node classification on a diverse array of heterophilic graphs to analyze the different performance trends of the low-pass filter. For each graph, separate experiments were conducted to study the effects of the addition of self-loops and parallel edges respectively. We applied GCN, a well-adopted low-pass filter, on 17 heterophilic graphs, and the respective performance trends are demonstrated in Tables 4, 5, and 6 for respectively standard heterophilic graphs, large-scale datasets, and currently proposed heterophilic graphs. The study reveals that each graph exhibits a steady pattern of either increasing or decreasing while adding either the self-loops or the parallel edges. We further marked the categories considering the performance trends for the increasing number of self-loops and parallel edges. For instance, GCN on Chameleon showed a decreasing trend while increasing the number of parallel edges, and on the contrary, performance increases on adding the parallel edges. Therefore, Chameleon was assumed to be in the category of A as per the rules from Table 1. Note that, the steady patterns are persistent across every dataset considered for the experimentation.

### 4.7 Observation from Performance Trends

The rationale against the backdrop of the performance trend can be explained through the lens of analyzing the spectrum of the graph. Since GCN performs the low-pass filtering, the coefficients of the lower frequencies will be enhanced, and the coefficients of the higher frequencies will be shrunk. Based on this, our discussion will revolve around analyzing 6 standard heterophilic graphs.

Table 6: GCN (a low-pass filter) is applied on the 5 newly-proposed heterophilic datasets curated by Platonov *et al.* (Platonov et al., 2023). Performance analyses are demonstrated after adding multiple self-loops and parallel edges respectively in the graphs. The performance trends are also highlighted and corresponding categories are marked for each dataset.

| Method | Input Adjacency | Roman-empire | Amazon-ratings | Minesweeper | Tolokers | Questions |
|---|---|---|---|---|---|---|
| GCN | A + I | $76.70 \pm 0.63$ | $42.01 \pm 0.58$ | $89.51 \pm 0.54$ | $80.10 \pm 1.11$ | $73.96 \pm 1.44$ |
| | A + 2I | $76.28 \pm 0.58$ | $41.58 \pm 0.56$ | $89.46 \pm 0.53$ | $80.13 \pm 1.06$ | $73.69 \pm 0.67$ |
| | A + 3I | $75.99 \pm 0.76$ | $41.56 \pm 0.53$ | $89.41 \pm 0.55$ | $80.03 \pm 1.02$ | $72.54 \pm 1.64$ |
| | A + 4I | $75.72 \pm 0.70$ | $41.49 \pm 0.40$ | $89.33 \pm 0.54$ | $79.76 \pm 1.02$ | $72.56 \pm 1.23$ |
| | A + 5I | $75.26 \pm 0.55$ | $41.21 \pm 0.60$ | $89.22 \pm 0.53$ | $79.70 \pm 0.98$ | $72.53 \pm 1.15$ |
| Trend | $\rightarrow$ | Decreasing ($\downarrow$) | Decreasing ($\downarrow$) | Decreasing ($\downarrow$) | Decreasing ($\downarrow$) | Decreasing ($\downarrow$) |
| GCN | A + I | $76.71 \pm 0.62$ | $42.00 \pm 0.58$ | $89.51 \pm 0.54$ | $80.10 \pm 1.12$ | $73.30 \pm 2.04$ |
| | 2A + I | $76.75 \pm 0.80$ | $41.93 \pm 0.50$ | $89.57 \pm 0.51$ | $79.95 \pm 1.08$ | $74.51 \pm 1.23$ |
| | 3A + I | $76.99 \pm 0.67$ | $41.93 \pm 0.90$ | $89.57 \pm 0.52$ | $79.87 \pm 0.93$ | $75.05 \pm 0.99$ |
| | 4A + I | $76.90 \pm 0.59$ | $42.02 \pm 0.53$ | $89.57 \pm 0.51$ | $79.87 \pm 0.99$ | $74.93 \pm 1.18$ |
| | 5A + I | $76.95 \pm 0.65$ | $42.18 \pm 0.77$ | $89.60 \pm 0.49$ | $79.75 \pm 0.94$ | $74.73 \pm 1.58$ |
| Trend | $\rightarrow$ | Increasing ($\uparrow$) | Increasing ($\uparrow$) | Increasing ($\uparrow$) | Decreasing ($\downarrow$) | Increasing ($\uparrow$) |
| Category | $\rightarrow$ | C | C | C | D | C |

**Addition of Self-loops** We observed that the addition of self-loops improves the performance of GCN on Cornell, Texas, Wisconsin, and Actor (Refer Table 4). Aligning to the theoretical analyses, adding self-loops will create a decreasing trend of the eigenvalues of $\tilde{L}$. Consequently, the number of lower frequencies will increase in the graph spectrum, which is beneficial for the performance boost of the GCN. Refer to Figure 4((a), (b), (c), (f)) to visualize the change of distribution of eigenvalues of $\tilde{L}$ with the addition of self-loops.

A deep observation reveals that the number of lower frequencies gradually increases while increasing the number of self-loops in the graph. This transformation offers a conducive environment for the operation of the low-pass filter (here GCN). Finally, the shape of the eigenvalue distribution resembles when $\alpha = 5$ for all four datasets.

A stark contrast was observed in the performance trends of the Chameleon and Squirrel datasets. Both of the datasets demonstrated degrading performance with the addition of an increasing number of self-loops. As mentioned earlier, the addition of self-loops will shrink the spectrum of the graph, leading to an increase in the lower frequencies. Refer 4((d), (e)) for the nuanced view of the alteration of eigenvalue distribution of $\tilde{L}$ for both datasets. Both histograms depict that the lower frequencies increase but eigenvalues are mostly centered towards 1 which might not be beneficial for the operation of GCN. Therefore, in this scenario, GCN witnessed deteriorating performance with the increasing number of self-loops.

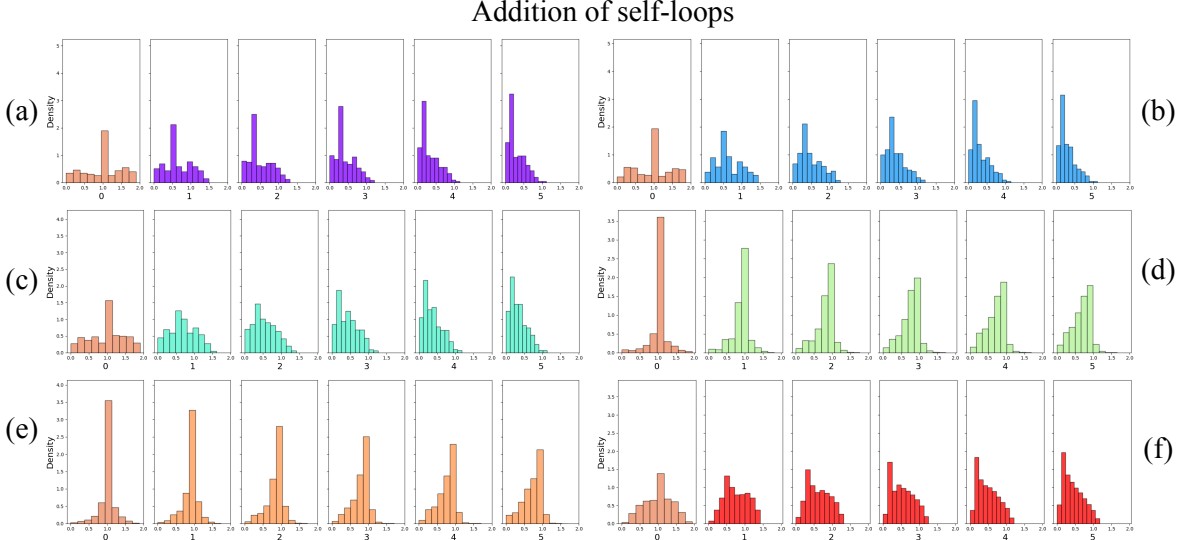

Addition of self-loops

Figure 4: The distribution of eigenvalues of normalized graph Laplacian for the datasets (a) Cornell, (b) Texas, (c) Wisconsin, (d) Chameleon, (e) Squirrel, and (f) Actor is demonstrated after adding self-loops. The initial distribution is shown on the left side of each diagram. The number of self-loops varies from 1 to 5 and corresponding changes are recorded.

**Addition of Parallel edges** The addition of parallel edges improves the performance of Cornell, Texas, Wisconsin, Chameleon, and Squirrel. The theoretical analysis illustrates that eigenvalues increase with the increasing number of parallel edges. Consequently, the number of higher frequencies will increase which seems to impede the performance of GCN. Refer to Figure 5((a), (b), (c), (d), (e)) to visualize the change in the distribution of eigenvalues of $\tilde{L}$ with the addition of the parallel edges. One common point should be mentioned that despite the increasing number of higher frequencies, many lower frequencies are still retained in the spectrum. This phenomenon performs the balance of the parity of lower and higher frequencies in the graph spectrum, eventually improving the performance of GCN.

On the contrary, GCN exhibited deteriorating performances on the Actor dataset with the increasing number of parallel edges. Refer to Figure 5(f) for the histogram asserting that the number of lower frequencies almost diminished. As a consequence, GCN witnessed a degrading performance.

**Key Takeaway** The crux of the experiments lies in the identification of trends (either increasing or decreasing) in the performance of low-pass filters with the addition of self-loops or parallel edges. Every input graph can be uniquely mapped to one of the four pre-defined categories depending on the performance trends. We marked the respective categories of all 17 heterophilic graphs involved in the experimentation. The assigned category characterizes the specific eigenvalue distribution of the normalized graph Laplacian for the corre-

sponding graph. The distribution of the eigenvalues offers significant insights into the structural patterns of the graphs like connected components, community structures, expansion properties, clustering, robustness, etc. The unraveling of such properties is often accomplished by pursuing computationally expensive eigenvalue decomposition or time-consuming prevailing graph algorithms. Amid the growing size of the graph resorting to such strategies leads to a labyrinth of the methodologies.

The size of the graphs like Cornell, Texas, or Chameleon is moderate and we performed eigenvalue decomposition to study the eigenvalue distribution of $\tilde{L}$. The initial distributions of the six graphs are presented in Figure 3. One can easily contemplate the high-level structural properties by observing the initial patterns. For instance, Cornell, Texas, and Wisconsin have a distribution mostly symmetrical around 1 in the spectrum, asserting the balanced parity of lower and higher frequencies. The observation underscores the presence of weakly connected components or isolated nodes. This is also validated from Table 3 as Cornell contains 47.5% isolated nodes. A similar argument applies to both Texas and Wisconsin. Conversely, Chameleon and Squirrel both have higher frequencies which signifies that graphs are dense, have strongly connected components, and contain no isolated nodes as confirmed from Table 3. In a different vein, Actor has an almost balanced parity of lower and higher frequencies characterizing the lower number of isolated nodes with a moderate average degree and density compared to the other five previously mentioned graphs.

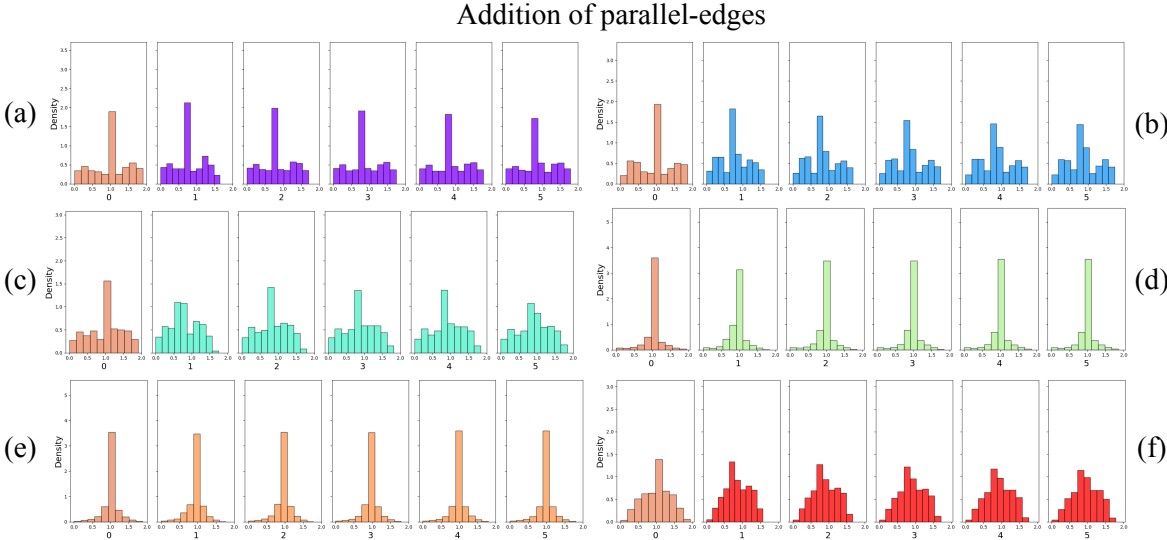

Figure 5: The distribution of eigenvalues of normalized graph Laplacian for the datasets (a) Cornell, (b) Texas, (c) Wisconsin, (d) Chameleon, (e) Squirrel, and (f) Actor are demonstrated after adding parallel edges. The initial distribution is shown on the left side of each diagram. The number of parallel edges varies from 1 to 5 and corresponding changes are recorded.

## 4.8   Inferring Characteristics of Spectrum for Large-scale Graphs

Evaluating the eigenvalue distribution of $\tilde{L}$ for a large-scale graph is critically challenging due to the potential computational overhead. Exploration of eigenvalue distribution is possible by separately observing the performance trends of the low-pass filters, with the addition of self-loops and parallel edges. This approach drastically reduces the computational budget and offers deeper insights into the intricate structural patterns in the given networks.

**arxiv-year, snap-patents, and genius** As per empirical evidence, GCN witnessed decreasing performance trends on arxiv-year, snap-patents, and genius in both cases of self-loops and parallel edges. The results emphasized that three networks contain a significantly lower number of non-zero frequencies with a substantial number of zero eigenvalues, asserting that the networks contain a large number of isolated nodes. The edge density and average degree of the graphs are also lower for the three graphs which can also be verified by

referring to Table 3. Therefore, we can predict the weak connectivity and the presence of isolated nodes in those graphs by only observing the performance trends with the addition of self-loops and parallel edges.

**Penn94** Penn94 was categorized in "C" resembling the category of Chameleon and Squirrel (Refer Figure 3). The initial frequency distribution of $\tilde{L}$ suggests a greater number of higher frequencies resulting in the densely connected network compared to the other five graphs (Refer Table 3). Also, Penn94 contains no isolated nodes sharing properties similar to those of Chameleon and Squirrel.

**twitch-gamers** The performance trends made twitch-gamers posited in category "B" which is identical to Actor. Thus, the eigenvalue distribution of twitch-gamers also resembles to Actor, having balanced centering to eigenvalue 1 and almost equal parity of lower and higher frequencies. Like Actor, twitch-gamers also have moderately dense and average node degrees with a lesser number of isolated nodes compared to the other five heterophilic graphs in this group.

**pokec** The category of pokec marked as $A$ shares identical characteristics with Cornell, Texas, and Wisconsin. The performance trend indicated the presence of the eigenvalues centering around eigenvalue 1 having balanced parity of lower and higher frequencies. Identically, pokec may contain isolated nodes and also weakly connected components.

Table 7: Analysis of the performance of GCN on Chameleon dataset is presented with the variation of number of self-loops and parallel edges simultaneously across the network.

| Dataset | # Parallel edges | | 1 | 2 | 3 | 4 | 5 |
|---|---|---|---|---|---|---|---|
| Chameleon | # Self-loops | 1 | $65.00 \pm 1.32$ | $68.61 \pm 1.52$ | $68.94 \pm 1.53$ | $69.14 \pm 1.35$ | $69.75 \pm 0.98$ |
| | | 2 | $59.84 \pm 1.97$ | $65.06 \pm 2.07$ | $66.71 \pm 1.35$ | $68.50 \pm 1.70$ | $68.64 \pm 1.77$ |
| | | 3 | $58.55 \pm 2.15$ | $68.61 \pm 2.26$ | $65.00 \pm 1.95$ | $66.88 \pm 2.01$ | $67.41 \pm 1.39$ |
| | | 4 | $58.04 \pm 2.78$ | $59.53 \pm 1.63$ | $62.82 \pm 2.03$ | $64.42 \pm 1.81$ | $67.03 \pm 1.90$ |
| | | 5 | $57.93 \pm 2.31$ | $59.16 \pm 1.54$ | $60.24 \pm 2.55$ | $63.70 \pm 2.00$ | $65.06 \pm 1.49$ |

**Roman-empire, Amazon-ratings, Minesweeper, and Questions** The four graphs have demonstrated declining performance trends while self-loops are added and performance is improved with the addition of parallel edges. The graphs thereby belonged to the category of "C" similar to Chameleon and Squirrel. This predicament has underscored that graphs contain higher frequencies than lower frequencies. The graph spectra are indicative of the higher sparsity and lower average degree of the graphs, compared to those of the Tolokers.

**Tolokers** Tolokers is categorized as "D" having similarity with arxiv-year, snap-patents, and genius. The performance trends signal the presence of a higher number of zero eigenvalues in the spectrum, containing a good number of isolated nodes (Refer to Table 3).

## 4.9 Runtime Comparison with Traditional Eigendecomposition Algorithms

We conducted an extensive study on 17 heterophilic benchmarks to compare the runtime of our proposed strategy with that of traditional eigendecomposition algorithms. We applied an inbuilt function `np.linalg.eig` implemented in the Numpy package to execute eigendecomposition. All experiments are carried out on a single 24 NVIDIA GeForce RTX 3090 GPU and we obtained the results as presented in Table 8. For each graph, we leverage 10 GNN training sessions for 5 self-loops and 5 parallel edges. For both cases, we utilized `time` function from Python. The analyses reveal that for small-scale graphs, the traditional algorithm outperforms our approach but for medium-sized graphs, our approach exhibited faster runtime in comparison to the inbuilt algorithm. Additionally, for large-scale graphs, our system showed Out-of-Memory (OOM) or process killed and was unable to estimate eigendecomposition. The experimental results underscore the significance of our approach to gain insights into the spectra without performing the time and memory-intensive existing eigendecomposition techniques.

Table 8: The comparative study between traditional eigendecomposition algorithms and our method applied to 17 heterophilic benchmarks. OOM denotes out-of-memory.

| Runtime (sec.) | Chameleon | Squirrel | Film | Texas | Cornell | Wisconsin |
|---|---|---|---|---|---|---|
| np.linalg.eig | 2.5334 | 22.1533 | 86.9033 | 0.0442 | 0.0226 | 0.0599 |
| Ours | 73.4782 | 162.0774 | 371.0116 | 52.9237 | 52.5499 | 52.9963 |

| Runtime (sec.) | arxiv-year | snap-patents | Penn94 | pokec | twitch-gamers | genius |
|---|---|---|---|---|---|---|
| np.linalg.eig | OOM | OOM | OOM | OOM | OOM | OOM |
| Ours | 365.9806 | 699.1057 | 2866.6932 | 1467.4539 | 1097.2305 | 576.4411 |

| Runtime (sec.) | Roman-empire | Amazon-ratings | Minesweper | Tolokers | Questions | |
|---|---|---|---|---|---|---|
| np.linalg.eig | 3240.4251 | 3032.0099 | 40.7877 | 94.4341 | process killed | |
| Ours | 95.5101 | 135.3722 | 79.7196 | 360.0264 | 247.7609 | |

## 4.10 Visualization of Spectrum for Heterophilic Graphs

We conducted a comparative study on the lower and higher frequencies of the six heterophilic graphs Chameleon, Squirrel, Texas, Cornell, Wisconsin, and Actor. The eigenvalue decomposition is performed on $\tilde{L}$ of individual datasets. The corresponding eigenvalues are divided into two sets $\lambda_{<1}$ and $\lambda_{\geq 1}$ as mentioned earlier. We estimated the percentage of the low frequencies and high frequencies of each dataset. Refer to Figure 6 for the detailed illustration. The figure delineates that the number of lower and higher

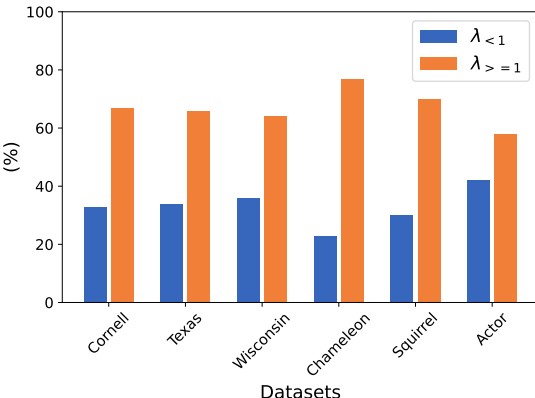

Figure 6: A comparative study on the number of low frequencies and high frequencies of the graph spectrum for six standard heterophilic graphs.

frequencies for Cornell, Texas, and Wisconsin are identical. In the experiments, their category is also similar which is "A". Concurrently, the Chameleon and Squirrel have similar patterns of the parity of eigenvalues and they also belong to a similar category "C". Furthermore, the pattern for the Actor is completely different from the rest of the others, and ends up marked as "B". The study established the unique interconnectedness of the parity of eigenvalues, frequency distribution of spectrum, and performance trends of low-pass filters.

## 4.11 Variation of both Self-loops and Parallel edges

A study is conducted to observe the effect on the performance of GCN with the variation of both self-loops and parallel edges. We consider the Chameleon as our candidate graph to serve the purpose. The number of self-loops and parallel edges both varied from 1 to 5, taking into account 25 combinations. GCN is applied to the modified Chameleon graph for each combination. The mean test accuracy with standard deviation estimated over 10 splits is reported against each combination. Refer to Table 7 to get a detailed illustration of the results. The uptick trend is noted for the increasing number of parallel edges in the network. While increasing the number of self-loops, a degradation in the model performance is observed. For a fixed number

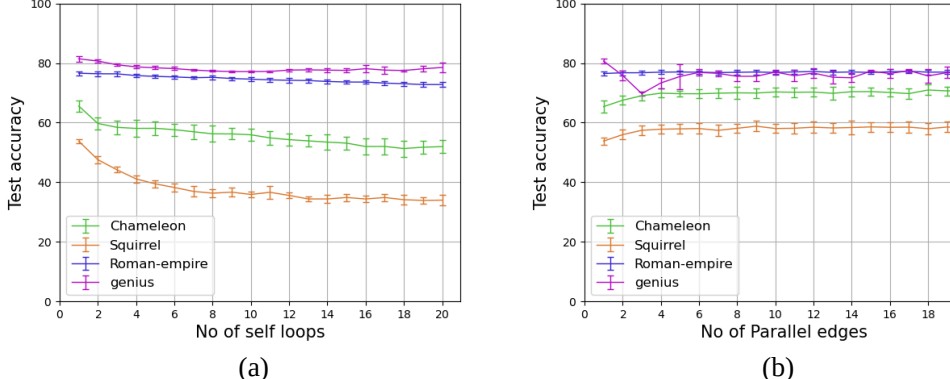

Figure 7: The effect on the performance of GCN on Chameleon, Squirrel, Roman-empire, and genius when (a) self-loops and (b) parallel edges are added a higher number of times is presented.

of self-loops, the test accuracy increases with the addition of parallel edges, irrespective of the beginning of the initial performance. Every combination of $(\alpha_i, \gamma_j)$ produces a filter with the adjusted lower and higher number of frequencies. Notably, the lowest performance is achieved when the number of self-loops is highest $(\alpha = 5)$ and the number of parallel edges is lowest $(\gamma = 1)$. The best performance is obtained with just the reverse settings like the lowest number of self-loops $(\alpha = 1)$ and the highest number of parallel edges $(\gamma = 5)$.

**Intuitive Explanation.** Adding self-loops increases the number of lower frequencies, shifting the graph spectrum towards the eigenvalue 0. Conversely, the addition of parallel edges increases the number of higher frequencies, shifting the spectrum toward the value 2. The addition of a maximum $\mathcal{P}$ number of self-loops and a maximum $\mathcal{Q}$ number of parallel edges shifts the graph spectrum to the left and right directions accordingly. Precisely, the addition of $\alpha \leq \mathcal{P}$-number of self-loops and $\gamma \leq \mathcal{Q}$-number of parallel edges will transform the graph spectrum intermediate between the two aforementioned extreme scenarios. The updated spectrum oscillation between the two extreme cases and the performance of GCN is also reflected in our quantitative analysis.

## 4.12   Effect on Performance with Very Large $\alpha$ and $\gamma$

We performed a study on the Chameleon, Squirrel, Roman-empire, and genius by increasing $\alpha$ and $\gamma$ to higher values to monitor the performance of GCN. The numbers of both $\alpha$ and $\gamma$ are varied from 1 to 20, and corresponding test accuracy with standard deviations are demonstrated in Figure 7. The study indicates the maintenance of the performance trends of the different datasets. The trend mostly depends on the input graph. For example, GCN on Chameleon showed a decrease (increase) in the number of self-loops (parallel edges). Conversely, GCN on genius exhibited a downtrend in performance by adding both self-loops and parallel edges. It is noteworthy that performance stabilized with the higher value of $\alpha$ or $\gamma$. Empirical observation points out that the change in eigenvalues will become comparably negligible when the value of $\alpha$ or $\gamma$ exceeds a certain limit. This phenomenon can be attributed to the saturating performance trends observed in the experiment.

## 4.13   Utility for Practitioners

In this work, we primarily focus on determining properties of the graph spectrum without resorting to costly eigenvalue decomposition. We map the input graph into one of the four categories based on the patterns observed in the performance of GCN. Beyond the theoretical insights and empirical evaluations, our work possesses some benefits for practitioners. We enjoy the following advantages without performing direct eigendecomposition.

- Our strategy reveals the shape of the graph spectrum characterizing structural properties like the presence of connected components, communities, etc. For example, regular graphs have a symmetric graph spectrum. Additionally, the spectral density can identify graph classes like scale-free, random, and small-world.

- Computing similarity measures between two large graphs is cumbersome. Our method offers insights into the spectrum distributions, whose comparisons enable quick similarity assessment between the large networks.

- We can detect anomalies of the evolving or dynamic networks by monitoring shifts in the respective distribution shifts.

## 5   Conclusion & Future Works

We conduct detailed studies regarding the performance of low-pass filters like GCN on the heterophilic graphs. The studies revealed that GCN showed monotone performance trends when the input graph is equipped with an increasing number of self-loops or parallel edges. We categorize the input graphs into four distinct categories based on these performance trends. We further observed that the eigenvalues of the normalized graph Laplacian decrease and increase when self-loops or parallel edges are added. Consequently, each category entails a significant amount of information pertaining to the characteristics of the graph spectrum. Therefore, the performance trends depend solely on the distribution of the eigenvalues in the entire spectrum. The graph spectrum patterns reveal the graph's intrinsic characteristics such as connected components, community structure, etc. Our work manifests a cost-effective pathway for estimating and understanding intrinsic properties and intricate patterns in the graph data, refraining from performing expensive computations. The design of effective application of GNNs to replace the prevailing costly algorithmic computations can be a potential avenue for future research directions.

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

# A    Appendix

## A.1    Proofs

This section will offer the detailed proofs and necessary derivations for Lemma 1, Lemma 2, Lemma 3, Lemma 4, Theorem 1, Theorem 2, Theorem 3, Theorem 4, and Corollary 2. Before delving into the detailed proofs and derivations, the following Lemma will assist in proving the following theorems. The following proposition is adopted from (Wu et al., 2019).

**Proposition 1.** *Let us assume $\beta_1 \leq \beta_2 \leq \cdots \leq \beta_n$ are the eigenvalues of $D^{-\frac{1}{2}}AD^{-\frac{1}{2}}$ and $\delta_1 \leq \delta_2 \leq \cdots \leq \delta_n$ are the eigenvalues of $\tilde{D}^{-\frac{1}{2}}A\tilde{D}^{-\frac{1}{2}}$ where $\tilde{D}_\alpha = D + \alpha I$, then we have the following inequalities*

$$\delta_1 \geq \frac{max_i d_i}{\alpha + max_i d_i}\beta_1, \qquad\qquad \delta_n \leq \frac{min_i d_i}{\alpha + min_i d_i}. \qquad (2)$$

*Proof.* Recall that $L_{\text{sym}} = I - D^{-\frac{1}{2}}AD^{-\frac{1}{2}}$ and a well-known fact is that 0 is an eigenvalue of $L_{\text{sym}}$. Therefore, we have $\beta_n = 1$. As $A$ is free of self-loops then $\text{Tr}(D^{-\frac{1}{2}}AD^{-\frac{1}{2}}) = 0 = \sum_i \beta_i$ which implies $\beta_1 < 0$.

Choose $x$ such that $||x|| = 1$ and consider $y = D^{\frac{1}{2}}\tilde{D}_\alpha^{-\frac{1}{2}}x$. Now, $||y||^2 = \sum_i \frac{d_i}{d_i + \alpha}x_i^2$. Also, we have $\frac{\min_i d_i}{\alpha + \min_i d_i} \leq ||y||^2 \leq \frac{\max_i d_i}{\alpha + \max_i d_i}$.

Applying the Rayleigh quotient, we have the following bound for the smallest eigenvalue $\alpha_1$,

$$
\begin{aligned}
\delta_1 &= \min_{||x||=1} (x^\top \tilde{D}_\alpha^{-\frac{1}{2}} A \tilde{D}_\alpha^{-\frac{1}{2}} x) \\
&= \min_{||x||=1} (y^\top D^{-\frac{1}{2}} A D^{-\frac{1}{2}} y) \qquad \text{(by variable substitution)} \\
&= \min_{||x||=1} (\frac{y^\top D^{-\frac{1}{2}} A D^{-\frac{1}{2}} y}{||y||^2} ||y||^2) \\
&\geq \min_{||x||=1} (\frac{y^\top D^{-\frac{1}{2}} A D^{-\frac{1}{2}} y}{||y||^2}) \max_{||x||=1} (||y||^2) \\
&(\because \min(f(z)g(z)) \geq \min(f(z)) \max(g(z)) \text{ if} \\
&\min(f(z)) < 0, \forall g(z) > 0 \\
&\text{and } \min_{||x||=1} (\frac{y^\top D^{-\frac{1}{2}} A D^{-\frac{1}{2}} y}{||y||^2}) = \beta_1 < 0) \\
&= \beta_1 \max_{||x||=1} ||y||^2 \\
&\geq \frac{\max_i d_i}{\alpha + \max_i d_i} \beta_1
\end{aligned}
\tag{3}
$$

Similarly, the upper bound for $\delta_n$ can be proved as $\delta_n \leq \frac{\min_i d_i}{\alpha + \min_i d_i}$. A similar problem can be solved for the parallel edge addition with $\tilde{D}_\gamma = (1 + \gamma)D + I$. and $\delta_1 = \min_{||x||=1}(x^\top \tilde{D}_\gamma^{-\frac{1}{2}} A \tilde{D}_\gamma^{-\frac{1}{2}} x)$. This problem will yield the bound as $\delta_1 \geq \frac{\max d_i}{1 + (1+\gamma)\max d_i} \beta_1$. The bound can be derived similarly as depicted in Eq. 3. $\square$

**Lemma 1** Consider a graph $G$ with $A$ and $D$ as the adjacency and degree matrix. Now $\alpha$-times self-loops are added in $G$ with $\alpha \in \mathbb{Z}^+$. Assume $\lambda_\alpha^{\max}$ is the maximum eigenvalue of symmetrically normalized graph Laplacian $\tilde{L}_\alpha$ of the updated graph. If $\beta_1$ is the smallest eigenvalue of $D^{-\frac{1}{2}} A D^{-\frac{1}{2}}$ and $\max_i d_i$ is the maximum degree of $G$, then $\lambda_\alpha^{\max} \leq \frac{\max_i d_i (1 - \beta_1)}{\alpha + \max_i d_i}$.

*Proof.* After applying $\alpha$-times self-loops the symmetrically normalized Laplacian is presented as:

$$
\begin{aligned}
\tilde{L}_\alpha &= I - \tilde{D}_\alpha^{-\frac{1}{2}} \tilde{A}_\alpha \tilde{D}_\alpha^{-\frac{1}{2}} \\
&= I - \tilde{D}_\alpha^{-\frac{1}{2}} (A + \alpha I) \tilde{D}_\alpha^{-\frac{1}{2}} \\
&= I - \tilde{D}_\alpha^{-\frac{1}{2}} A \tilde{D}_\alpha^{-\frac{1}{2}} - \alpha \tilde{D}_\alpha^{-1}
\end{aligned}
\tag{4}
$$

Let $\lambda_\alpha^{\max}$ is the maximum eigenvalue of $\tilde{L}_\alpha$ and applying Rayleigh quotient the following can be obtained

$$
\begin{aligned}
\lambda_\alpha^{\max} &= \max_{||x||=1} x^\top \tilde{L}_\alpha x \\
&= \max_{||x||=1} x^\top (I - \tilde{D}_\alpha^{-\frac{1}{2}} A \tilde{D}_\alpha^{-\frac{1}{2}} - \alpha \tilde{D}_\alpha^{-1}) x \\
&\leq (1 - \min_{||x||=1} x^\top \tilde{D}_\alpha^{-\frac{1}{2}} A \tilde{D}_\alpha^{-\frac{1}{2}} x - \min_{||x||=1} \alpha x^\top \tilde{D}_\alpha^{-1} x) \\
&= 1 - \delta_1 - \frac{\alpha}{\alpha + \max_i d_i} \qquad \text{(from Proposition 1)} \\
&\leq 1 - \frac{\max_i d_i}{\alpha + \max_i d_i} \beta_1 - \frac{\alpha}{\alpha + \max_i d_i} \\
&= \frac{\max_i d_i (1 - \beta_1)}{\alpha + \max_i d_i}
\end{aligned}
\tag{5}
$$

When $\alpha$ increases the upper bound of $\lambda_\alpha^{\max}$ decreases which indicates the possible shrinking of the maximum eigenvalue of the graph spectrum.

$\square$

**Lemma 2** Consider a graph $G$ with $A$ and $D$ as the adjacency and degree matrix. Now $\gamma$-times parallel edges are added in $G$ with $\gamma \in \mathbb{Z}^+$. Assume $\lambda_\gamma^{\max}$ is the maximum eigenvalue of symmetrically normalized graph Laplacian $\tilde{L}_\gamma$ of the updated graph. If $\beta_1$ is the smallest eigenvalue of $D^{-\frac{1}{2}}AD^{-\frac{1}{2}}$ and $\max_i d_i$ is the maximum degree of $G$, then $\lambda_\gamma^{\max} \leq \frac{(1+\gamma)\max_i d_i(1-\beta_1)}{1+(1+\gamma)\max_i d_i}$.

*Proof.* After applying $\gamma$-times parallel edges in the graph, the symmetrically normalized graph Laplacian will be

$$
\begin{aligned}
\tilde{L}_\gamma &= I - \tilde{D}_\gamma^{-\frac{1}{2}}\tilde{A}_\gamma \tilde{D}_\gamma^{-\frac{1}{2}} \\
&= I - \tilde{D}_\gamma^{-\frac{1}{2}}((\gamma+1)A + I)\tilde{D}_\gamma^{-\frac{1}{2}} \\
&= I - (\gamma+1)\tilde{D}_\gamma^{-\frac{1}{2}}A\tilde{D}_\gamma^{-\frac{1}{2}} - \tilde{D}_\gamma^{-1}
\end{aligned}
\tag{6}
$$

Let $\lambda_\gamma^{\max}$ is the maximum eigenvalue of $\tilde{L}_\gamma$ and applying Rayleigh quotient the following can be obtained

$$
\begin{aligned}
\lambda_\gamma^{\max} &= \max_{||x||=1} x^\top \tilde{L}_\gamma x \\
&= \max_{||x||=1} x^\top (I - (\gamma+1)\tilde{D}_\gamma^{-\frac{1}{2}}A\tilde{D}_\gamma^{-\frac{1}{2}} - \tilde{D}_\gamma^{-1})x \\
&\leq (1 - (\gamma+1)\min_{||x||=1} x^\top \tilde{D}_\gamma^{-\frac{1}{2}}A\tilde{D}_\gamma^{-\frac{1}{2}}x - \min_{||x||=1} x^\top \tilde{D}_\gamma^{-1}x) \\
&= 1 - (\gamma+1)\delta_1 - \frac{1}{1+(1+\gamma)\max_i d_i} \quad \text{(from Proposition 1)} \\
&\leq 1 - \frac{(\gamma+1)\max_i d_i}{1+(1+\gamma)\max_i d_i}\beta_1 - \frac{1}{1+(1+\gamma)\max_i d_i} \\
&= \frac{(1+\gamma)\max_i d_i(1-\beta_1)}{1+(1+\gamma)\max_i d_i}
\end{aligned}
\tag{7}
$$

When $\gamma$ increases, the upper bound of $\lambda_\gamma^{\max}$ increases which shows the possible expansion of the maximum eigenvalue of the graph spectrum.

$\square$

**Lemma 3** Given a $k$-regular graph $\mathcal{G}$, the eigenvalues of $\tilde{A}_N^\alpha$ will lie in $[-1, 1]$ $\forall \alpha \geq 1$.

*Proof.* Consider a $k$-regular graph $\mathcal{G}$ where each node has a degree $k$ with the normalized adjacency matrix is $\tilde{A}_N = \tilde{D}^{-\frac{1}{2}}\tilde{A}\tilde{D}^{-\frac{1}{2}}$ where $\tilde{A} = A + I, \tilde{D} = D + I$. If $\alpha$-times (with $\alpha \geq 1$) self-loops are added, then the updated normalized adjacency matrix is $\tilde{A}_N^\alpha = \tilde{D}_\alpha^{-\frac{1}{2}}\tilde{A}_\alpha \tilde{D}_\alpha^{-\frac{1}{2}}$ where $\tilde{A}_\alpha = A + \alpha I, \tilde{D}_\alpha = D + \alpha I$. As the graph is regular then $\tilde{A}_N$ can be presented as:

$$
\begin{aligned}
\tilde{A}_N &= \tilde{D}^{-\frac{1}{2}}\tilde{A}\tilde{D}^{-\frac{1}{2}} \\
&= \frac{1}{\sqrt{k+1}}(A+I)\frac{1}{\sqrt{k+1}} \\
&= \frac{1}{k+1}(A+I)
\end{aligned}
\tag{8}
$$

In a similar fashion, we can express $\tilde{A}_N^\alpha = \frac{1}{k+\alpha}(A + \alpha I)$. Since, both $\tilde{A}_N$ and $\tilde{A}_N^\alpha$ are the linearly scaled transformations of $A$, then both will share a similar set of eigenvectors with different scaled eigenvalues. Assume $v$ is an eigenvector of $\tilde{A}_N$ associated with any eigenvalue $\lambda_1$. Therefore,

$$
\begin{aligned}
\tilde{A}_N v &= \lambda_1 v \\
\frac{1}{k+1}(A+I)v &= \lambda_1 v \\
Av &= ((k+1)\lambda_1 - 1)v
\end{aligned}
\tag{9}
$$

We can say that $v$ is an eigenvector of $A$ with corresponding eigenvalue $\lambda_A = ((k+1)\lambda_1 - 1)$. Consider the eigenvector $v$ of $\tilde{A}_N^\gamma$ with the with the eigenvalue $\lambda_2$. Then, we have the following,

$$
\begin{aligned}
\tilde{A}_N^\alpha v &= \lambda_2 v \\
\frac{1}{k+\alpha}(A + \alpha I)v &= \lambda_2 v \\
\lambda_2 &= \frac{\lambda_A + \alpha}{k + \alpha}
\end{aligned}
\tag{10}
$$

As the range of eigenvalues of $\tilde{A}_N$ is $[-1, 1]$, thus we have $\lambda_1 \leq 1$. The following inequality can be expressed,

$$
\begin{aligned}
\lambda_1 &\leq 1 \\
(k+1)\lambda_1 &\leq k + 1 \\
(k+1)\lambda_1 - 1 &\leq k + 1 - 1 \\
\lambda_A &\leq k
\end{aligned}
\tag{11}
$$

Using the inequality we will prove the next stage as

$$
\begin{aligned}
\lambda_A &\leq k \\
\alpha + \lambda_A &\leq \alpha + k \\
\frac{\alpha + \lambda_A}{k + \alpha} &\leq \frac{\alpha + k}{\alpha + k} \\
\lambda_2 &\leq 1
\end{aligned}
\tag{12}
$$

In this way, we showed that any eigenvalue of $\tilde{A}_N^\alpha$ is 1. For the lower bound, we can write,

$$
\begin{aligned}
\lambda_1 &\geq -1 \\
(k+1)\lambda_1 &\geq -k - 1 \\
(k+1)\lambda_1 - 1 &\geq -k - 1 - 1 \\
\lambda_A &\geq -k - 2
\end{aligned}
\tag{13}
$$

Using the inequality we will prove the next stage as,

$$
\begin{aligned}
\lambda_A &\geq -k - 2 \\
\alpha + \lambda_A &\geq \alpha - k - 2 \\
\frac{\alpha + \lambda_A}{k + \alpha} &\geq \frac{\alpha - k - 2}{\alpha + k} \\
\lambda_2 &\geq 1 - \frac{2(k+1)}{k + \alpha}
\end{aligned}
\tag{14}
$$

The degree $k > 1$, then we have $\lambda_2 \geq -1$. Therefore, the addition of self-loops will not alter the range of the eigenvalues of the symmetrically normalized adjacency matrix. $\qquad\square$

**Lemma 4** Given a $k$-regular graph $\mathcal{G}$, the eigenvalues of $\tilde{A}_N^\gamma$ will lie in $[-1, 1]\ \forall \gamma \geq 1$.

*Proof.* Consider a $k$-regular graph $\mathcal{G}$ where each node has a degree $k$ with the normalized adjacency matrix is $\tilde{A}_N = \tilde{D}^{-\frac{1}{2}}\tilde{A}\tilde{D}^{-\frac{1}{2}}$ where $\tilde{A} = A + I, \tilde{D} = D + I$. If $\gamma$-times (with $\gamma \geq 1$) parallel edges are added, then the updated normalized adjacency matrix is $\tilde{A}_N^\gamma = \tilde{D}_\gamma^{-\frac{1}{2}}\tilde{A}_\gamma \tilde{D}_\gamma^{-\frac{1}{2}}$ where $\tilde{A}_\gamma = (1 + \gamma)A + I, \tilde{D}_\gamma = (1 + \gamma)D + I$. As the graph is regular then $A_N$ can be presented as:

$$
\begin{aligned}
\tilde{A}_N &= \tilde{D}^{-\frac{1}{2}}\tilde{A}\tilde{D}^{-\frac{1}{2}} \\
&= \frac{1}{\sqrt{k+1}}(A + I)\frac{1}{\sqrt{k+1}} \\
&= \frac{1}{k+1}(A + I)
\end{aligned}
\tag{15}
$$

In a similar fashion, we can express $\tilde{A}_N^\gamma = \frac{1}{1+(1+\gamma)k}((1+\gamma)A+I)$. Since, both $\tilde{A}_N$ and $\tilde{A}_N^\gamma$ are the linearly scaled transformations of $A$, then both will share a similar set of eigenvectors with different scaled eigenvalues. Assume $v$ is an eigenvector of $\tilde{A}_N$ associated with any eigenvalue $\lambda_1$. Therefore,

$$\tilde{A}_N v = \lambda_1 v$$
$$\frac{1}{k+1}(A+I)v = \lambda_1 v \tag{16}$$
$$Av = ((k+1)\lambda_1 - 1)v$$

We can say that $v$ is an eigenvector of $A$ with corresponding eigenvalue $\lambda_A = ((k+1)\lambda_1 - 1)$. Consider the eigenvector $v$ of $\tilde{A}_N^\gamma$ with the with the eigenvalue $\lambda_2$. Then, we have the following,

$$\tilde{A}_N^\gamma v = \lambda_2 v$$
$$\frac{1}{1+(1+\gamma)k}((1+\gamma)A+I)v = \lambda_2 v \tag{17}$$
$$\lambda_2 = \frac{(1+\gamma)\lambda_A + 1}{(1+\gamma)k + 1}$$

As the range of eigenvalues of $\tilde{A}_N$ is $[-1, 1]$, thus we have $\lambda_1 \leq 1$. The following inequality can be expressed,

$$\lambda_1 \leq 1$$
$$(k+1)\lambda_1 \leq k+1$$
$$(k+1)\lambda_1 - 1 \leq k+1-1 \tag{18}$$
$$\lambda_A \leq k$$

Using the inequality we will prove the next stage as

$$\lambda_A \leq k$$
$$(1+\gamma)\lambda_A \leq (1+\gamma)k$$
$$(1+\gamma)\lambda_A \leq (1+\gamma)k$$
$$(1+\gamma)\lambda_A + 1 \leq (1+\gamma)k + 1 \tag{19}$$
$$\frac{(1+\gamma)\lambda_A + 1}{(1+\gamma)k + 1} \leq 1$$
$$\lambda_2 \leq 1$$

In this way, we have shown the maximum eigenvalue of $\tilde{A}_N^\gamma$ is 1. The lower bound of the eigenvalues can be shown as

$$\lambda_1 \geq -1$$
$$(k+1)\lambda_1 \geq -k-1$$
$$(k+1)\lambda_1 - 1 \geq -k-1-1 \tag{20}$$
$$\lambda_A \leq -k-2$$

Using the inequality we will prove the next stage as

$$\lambda_A \leq -k-2$$
$$(1+\gamma)\lambda_A \geq (1+\gamma)(-k-2)$$
$$(1+\gamma)\lambda_A + 1 \geq (1+\gamma)(-k-2) + 1$$
$$\frac{(1+\gamma)\lambda_A + 1}{(1+\gamma)k + 1} \geq \frac{(1+\gamma)(-k-2)+1}{(1+\gamma)k + 1} \tag{21}$$
$$\lambda_2 \geq 1 - \frac{2(k+1)(\gamma+1)}{(1+\gamma)k + 1}$$

As $k, \gamma > 0$, then we have $\lambda_2 \geq -1$. Therefore, the addition of parallel edges will not alter the range of the eigenvalues of the symmetrically normalized adjacency matrix. $\square$

**Theorem 1** Consider a $k$-regular graph with $\alpha_1, \alpha_2 \in \mathbb{R}^+$ with $\alpha_1 \leq \alpha_2$, then $\lambda_{\alpha_1}^i \geq \lambda_{\alpha_2}^i, \forall\, 1 \leq i \leq n$, where $\lambda_{\alpha_1}^i$ and $\lambda_{\alpha_2}^i$ are the $i^{th}$ eigenvalues of $\tilde{L}_{\alpha_1}$ and $\tilde{L}_{\alpha_2}$ respectively.

*Proof.* Consider a $k$-regular graph $\mathcal{G}$ where each node of degree $k$ with the normalized adjacency matrix is $A_N = \tilde{D}^{-\frac{1}{2}} \tilde{A} \tilde{D}^{-\frac{1}{2}}$ where $\tilde{A} = A + I, \tilde{D} = D + I$. If $\alpha$-times (with $\alpha \geq 1$) self-loops are added, then the updated normalized adjacency matrix is $\tilde{A}_N^\alpha = \tilde{D}_\alpha^{-\frac{1}{2}} \tilde{A}_\alpha \tilde{D}_\alpha^{-\frac{1}{2}}$ where $\tilde{A}_\alpha = A + \alpha I, \tilde{D}_\alpha = D + \alpha I$. As the graph is regular then we can have the following

$$
\begin{aligned}
A_N^\alpha &= \tilde{D}_\alpha^{-\frac{1}{2}} \tilde{A}_\alpha \tilde{D}_\alpha^{-\frac{1}{2}} \\
&= \frac{1}{\sqrt{k+\alpha}} (A + \alpha I) \frac{1}{\sqrt{k+\alpha}} \\
&= \frac{1}{(k+\alpha)} (A + \alpha I)
\end{aligned}
\tag{22}
$$

Therefore, we can express $A_N^{\alpha_1} = \frac{1}{k+\alpha_1}(A + \alpha_1 I)$ and $A_N^{\alpha_2} = \frac{1}{k+\alpha_2}((A + \alpha_2 I)$. Since, $A_N^{\alpha_1}$ and $A_N^{\alpha_2}$ are the linear transformation of $A$, both will have the same set of eigenvectors but with different eigenvalues. Assume $v$ is the eigenvector of $A$ and its corresponding eigenvalue is $\lambda_{\alpha_1}$. Therefore, the following can be written as

$$
\begin{aligned}
A_N^{\alpha_1} v &= \lambda_{\alpha_1} v \\
\frac{1}{(k+\alpha_1)} (A + \alpha_1 I) v &= \lambda_{\alpha_1} v \\
A v &= ((k+\alpha_1)\lambda_{\alpha_1} - \alpha_1) v
\end{aligned}
\tag{23}
$$

We can say that $v$ is the eigenvector of $A$ with the corresponding eigenvalue $\lambda_A = ((k+\alpha_1)\lambda_{\alpha_1} - \alpha_1)$. For $A_N^{\alpha_2}$, for eigenvector $v$, the corresponding eigenvalue is $\lambda_{\alpha_2}$. Now, we have the following:

$$
\begin{aligned}
A_N^{\alpha_2} v &= \lambda_{\alpha_2} v \\
\frac{1}{(k+\alpha_2)} (A + \alpha_2 I) v &= \lambda_{\alpha_2} v \\
A v &= ((k+\alpha_2)\lambda_{\alpha_2} - \alpha_2) v
\end{aligned}
\tag{24}
$$

Similarly, we can also express $\lambda_A = ((k+\alpha_2)\lambda_{\alpha_2} - \alpha_2)$. Now, equating the two different expressions of $\lambda_A$, we can express the following

$$
\begin{aligned}
&(k+\alpha_1)\lambda_{\alpha_1} - \alpha_1 = (k+\alpha_2)\lambda_{\alpha_2} - \alpha_2 \\
&k\lambda_{\alpha_1} - \alpha_1(1 - \lambda_{\alpha_1}) = k\lambda_{\alpha_2} - \alpha_2(1 - \lambda_{\alpha_2}) \\
&k(\lambda_{\alpha_1} - \lambda_{\alpha_2}) = \alpha_1(1 - \lambda_{\alpha_1}) - \alpha_2(1 - \lambda_{\alpha_2}) \\
&\text{As per provided condition, we have } \alpha_1 < \alpha_2 \\
&\alpha_1(1 - \lambda_{\alpha_1}) < \alpha_2(1 - \lambda_{\alpha_1}) \\
&\alpha_1(1 - \lambda_{\alpha_1}) - \alpha_2(1 - \lambda_{\alpha_2}) < \alpha_2(1 - \lambda_{\alpha_1}) - \alpha_2(1 - \lambda_{\alpha_2}) \\
&k(\lambda_{\alpha_1} - \lambda_{\alpha_2}) < \alpha_2(1 - \lambda_{\alpha_1}) - \alpha_2(1 - \lambda_{\alpha_2}) \\
&k(\lambda_{\alpha_1} - \lambda_{\alpha_2}) < \alpha_2(\lambda_{\alpha_2} - \lambda_{\alpha_1}) \\
&(k+\alpha_2)(\lambda_{\alpha_!} - \lambda_{\alpha_2}) < 0 \\
&\text{As } k, \alpha_2 > 0 \text{ then} \\
&\lambda_{\alpha_!} - \lambda_{\alpha_2} < 0 \\
&\lambda_{\alpha_!} < \lambda_{\alpha_2}
\end{aligned}
$$

The above equation holds for all $|\lambda_{\alpha_1}|, |\lambda_{\alpha_2}| \leq 1$ which is ensured from Lemma 3. The eigenvalue of $A_N^{\alpha_2}$ became greater than that of $A_N^{\alpha_1}$ when self-loops are added to the graph. We know that $\tilde{L}_\alpha = \mathrm{I} - \tilde{D}_\alpha^{-\frac{1}{2}} \tilde{A}_\alpha \tilde{D}_\alpha^{-\frac{1}{2}}$, indicates if the eigenvalue of $A_N^\alpha$ increases then the corresponding eigenvalue of $\tilde{L}_\alpha$ decreases. Therefore, it can be concluded that the eigenvalue of $\tilde{L}_\alpha$ decreases with the addition of self-loops. $\qquad\square$

**Theorem 2** Consider a $k$-regular graph with $\gamma_1, \gamma_2 \in \mathbb{R}^+$ with $\gamma_1 \leq \gamma_2$, then $\lambda_{\gamma_1}^i \leq \lambda_{\gamma_2}^i, \forall\ 1 \leq i \leq n$, where $\lambda_{\gamma_1}^i$ and $\lambda_{\gamma_2}^i$ are the $i^{th}$ eigenvalues of $\tilde{L}_{\gamma_1}$ and $\tilde{L}_{\gamma_2}$ respectively.

*Proof.* Consider a $k$-regular graph $\mathcal{G}$ where each node has a degree $k$ with the normalized adjacency matrix is $A_N = \tilde{D}^{-\frac{1}{2}}\tilde{A}\tilde{D}^{-\frac{1}{2}}$ where $\tilde{A} = A + I, \tilde{D} = D + I$. If $\gamma$-times (with $\gamma \geq 1$) parallel edges are added, then the updated normalized adjacency matrix is $\tilde{A}_N^\gamma = \tilde{D}_\gamma^{-\frac{1}{2}}\tilde{A}_\gamma\tilde{D}_\gamma^{-\frac{1}{2}}$ where $\tilde{A}_\gamma = (1+\gamma)A + I, \tilde{D}_\gamma = (1+\gamma)D + I$. As the graph is regular then we can have the following

$$
\begin{aligned}
A_N^\gamma &= \tilde{D}_\gamma^{-\frac{1}{2}}\tilde{A}_\gamma\tilde{D}_\gamma^{-\frac{1}{2}} \\
&= \frac{1}{\sqrt{1+(1+\gamma)k}}((1+\gamma)A + I)\frac{1}{\sqrt{1+(\gamma+1)k}} \\
&= \frac{1}{1+(1+\gamma)k}((1+\gamma)A + I)
\end{aligned}
\tag{25}
$$

Therefore, we can express $A_N^{\gamma_1} = \frac{1}{1+(1+\gamma_1)k}((1+\gamma_1)A + I)$ and $A_N^{\gamma_2} = \frac{1}{1+(1+\gamma_2)k}((1+\gamma_2)A + I)$. Since, $A_N^{\gamma_1}$ and $A_N^{\gamma_2}$ are the scalar transformation of $A$, both will have the same set of eigenvectors with different eigenvalues. Assume $v$ is the eigenvector of $A$ and its corresponding eigenvalue is $\lambda_{\gamma_1}$. Therefore, the following can be written as

$$
\begin{aligned}
A_N^{\gamma_1}v &= \lambda_{\gamma_1}v \\
\frac{1}{1+(1+\gamma_1)k}((1+\gamma_1)A + I)v &= \lambda_{\gamma_1}v \\
Av &= \frac{(1+(1+\gamma_1)k)\lambda_{\gamma_1} - 1}{1+\gamma_1}v
\end{aligned}
\tag{26}
$$

We can say that $v$ is the eigenvector of $A$ with the corresponding eigenvalue $\lambda_A = \frac{(1+(1+\gamma_1)k)\lambda_{\gamma_1}-1}{1+\gamma_1}$. For $A_N^\gamma$, the eigenvector is $v$ with the eigenvalue $\lambda_2$. Then, we have the following

$$
\begin{aligned}
A_N^{\gamma_2}v &= \lambda_{\gamma_2}v \\
\frac{1}{1+(1+\gamma_2)k}((1+\gamma_2)A + I)v &= \lambda_{\gamma_2}v \\
Av &= \frac{(1+(1+\gamma_2)k)\lambda_{\gamma_2} - 1}{1+\gamma_2}v
\end{aligned}
\tag{27}
$$

We can also express $\lambda_A = \frac{(1+(1+\gamma_2)k)\lambda_{\gamma_2}-1}{1+\gamma_2}$. Now, equating the two different values of $\lambda_A$, we can have the following

$$
\begin{aligned}
\frac{(1+(1+\gamma_1)k)\lambda_{\gamma_1}-1}{1+\gamma_1} &= \frac{(1+(1+\gamma_2)k)\lambda_{\gamma_2}-1}{1+\gamma_2}\\
(1+\gamma_2)((1+(1+\gamma_1)k)\lambda_{\gamma_1}-1) &=\\
&\quad (1+\gamma_1)(1+(1+\gamma_2)k)\lambda_{\gamma_2}-1)\\
(1+\gamma_2)(1+(1+\gamma_1)k)\lambda_{\gamma_1}-(1+\gamma_2) &=\\
&\quad (1+\gamma_1)(1+(1+\gamma_2)k)\lambda_{\gamma_2}-(1+\gamma_1)\\
(1+\gamma_2+(1+\gamma_1)(1+\gamma_2)k)\lambda_{\gamma_1}-1-\gamma_2 &=\\
&\quad (1+\gamma_1+(1+\gamma_1)(1+\gamma_2)k)\lambda_{\gamma_2}-1-\gamma_1\\
(1+\gamma_2+(1+\gamma_1)(1+\gamma_2)k)\lambda_{\gamma_1} &=\\
&\quad (\gamma_2-\gamma_1)+(1+\gamma_1+(1+\gamma_1)(1+\gamma_2)k)\lambda_{\gamma_2}\\
\lambda_{\gamma_1} =\frac{\gamma_2-\gamma_1}{(1+\gamma_2+(1+\gamma_1)(1+\gamma_2)k)} &+\\
&\quad \frac{(1+\gamma_1+(1+\gamma_1)(1+\gamma_2)k)}{(1+\gamma_2+(1+\gamma_1)(1+\gamma_2)k)}\lambda_{\gamma_2}\\
=\frac{\gamma_2-\gamma_1}{(1+\gamma_2+(1+\gamma_1)(1+\gamma_2)k)} &+\\
&\quad \frac{(1+\gamma_2+(1+\gamma_1)(1+\gamma_2)k)-\gamma_2+\gamma_1}{(1+\gamma_2+(1+\gamma_1)(1+\gamma_2)k)}\lambda_{\gamma_2}\\
=\frac{\gamma_2-\gamma_1}{(1+\gamma_2+(1+\gamma_1)(1+\gamma_2)k)} &+\\
&\quad (1-\frac{\gamma_2-\gamma_1}{(1+\gamma_2+(1+\gamma_1)(1+\gamma_2)k)})\lambda_{\gamma_2}\\
=\frac{\gamma_2-\gamma_1}{(1+\gamma_2+(1+\gamma_1)(1+\gamma_2)k)}(1-\lambda_{\gamma_2}) &+\lambda_{\gamma_2}
\end{aligned}
\tag{28}
$$

According to the condition provided $\gamma_2 > \gamma_1$ we can say

$$\lambda_{\gamma_1} > \lambda_{\gamma_2}$$

The eigenvalue of $A_N^{\gamma_2}$ became lesser than that of $A_N^{\gamma_1}$ with the addition of parallel edges. The Eq. 28 holds for $|\lambda_{\gamma_2}| \leq 1$ which is assured from Lemma 4. We know that $\tilde{L}_\gamma = I - \tilde{D}_\gamma^{-\frac{1}{2}}\tilde{A}_\gamma\tilde{D}_\gamma^{-\frac{1}{2}}$, indicates if the eigenvalue of $A_N^\gamma$ decreases then the corresponding eigenvalue of $\tilde{L}_\gamma$ increases. Therefore, it can be concluded that the eigenvalue of $\tilde{L}_\gamma$ increases with the addition of parallel edges for the regular graphs. $\qquad\square$

**Corollary 2.** *The increase in the eigenvalues of $L_\gamma$ is independent of the number of self-loop additions in $\mathcal{G}$. On the contrary, The eigenvalues of $\tilde{L}_\gamma$ will increase if at least one self-loop is added per node in $\mathcal{G}$.*

*Proof.* Let us prove the statement by contradiction. We know $L = D - A$ and after adding $\alpha$-times self-loops and $\gamma$-times parallel edges, the $L_\gamma = D_\gamma - A_\gamma$ where $\tilde{A}_\gamma = (1+\gamma)A + \alpha I, \tilde{D}_\gamma = (\gamma+1)D + \alpha I$. Then,

$$
\begin{aligned}
L_\gamma &= (\tilde{D}_\gamma - \tilde{A}_\gamma)\\
&= (((\gamma+1)D + \alpha I) - ((\gamma+1)A + \alpha I))\\
&= (\gamma+1)(D-A)\\
&= (\gamma+1)L
\end{aligned}
\tag{29}
$$

The equation is independent of the number of self-loops we confirm that the effect of adding parallel edges prevails. Similarly, let us also prove the second part by contradiction. We know that $\tilde{L}_\gamma = I - \tilde{D}_\gamma^{-\frac{1}{2}}\tilde{A}_\gamma\tilde{D}_\gamma^{-\frac{1}{2}}$ and consider the expression without self-loops with the addition of $\gamma$-times parallel edges as $\tilde{D} = (1+\gamma)D, \tilde{A} =$

$(1 + \gamma)A$.

$$
\begin{aligned}
\tilde{L}_\gamma &= \mathrm{I} - \tilde{D}_\gamma^{-\frac{1}{2}} \tilde{A}_\gamma \tilde{D}_\gamma^{-\frac{1}{2}} \\
&= \mathrm{I} - \frac{1}{\sqrt{(1 + \gamma)}} D^{-\frac{1}{2}} (1 + \gamma) A \frac{1}{\sqrt{(1 + \gamma)}} D^{-\frac{1}{2}} \\
&= \mathrm{I} - D^{-\frac{1}{2}} A D^{-\frac{1}{2}} \\
&= \tilde{L}
\end{aligned}
\tag{30}
$$

Therefore, adding parallel edges without at least one self-loop per node does not change the normalized graph Laplacian. Hence, the result is proved. $\qquad\square$

**Theorem 3** Consider a connected graph $\mathcal{G}$ with $A_N = D^{-\frac{1}{2}} A D^{-\frac{1}{2}}$. Assuming the diagonal of $A$ of $G$ is perturbed by a significantly small $\alpha > 0$, then the updated normalized adjacency matrix will be $A_N^\alpha$. The change in the eigenvalues of $A_N^\alpha$ with respect to eigenvalues of $A_N$ will increase when $\alpha$ increases.

*Proof.* If the diagonal of $A$ is perturbed by $\alpha$, the symmetrically normalized graph Laplacian of $G$ will be,

$$
\tilde{L}_\alpha = \mathrm{I} - \tilde{D}_\alpha^{-\frac{1}{2}} \tilde{A}_\alpha \tilde{D}_\alpha^{-\frac{1}{2}},
\tag{31}
$$

where $\tilde{A}_\alpha = A + \alpha \mathrm{I}$ and $\tilde{D}_\alpha = D + \alpha \mathrm{I}$. Let us denote the normalized adjacency matrix without self-loops as $A_N = D^{-\frac{1}{2}} A D^{-\frac{1}{2}}$. The element of $A_N$ is represented as:

$$
A_N[i][j] = \begin{cases} \frac{A_{ij}}{\sqrt{d_i d_j}}, & i \neq j \\ 0, & i = j \end{cases}
\tag{32}
$$

After perturbed by $\alpha$, the normalized adjacency will be $A_N^\alpha = \tilde{D}_\alpha^{-\frac{1}{2}} \tilde{A}_\alpha \tilde{D}_\alpha^{-\frac{1}{2}}$. The elements of $\tilde{A}_N^\alpha$ can be represented as:

$$
A_N^\alpha[i][j] = \begin{cases} \frac{A_{ij}}{\sqrt{\alpha + d_i} \sqrt{\alpha + d_j}} & i \neq j \\ \frac{\alpha}{\alpha + d_i} & i = j \end{cases}
\tag{33}
$$

The entry-wise change in the normalized adjacency matrix is presented as:

$$
\delta A_N[i][j] = \begin{cases} \frac{A_{ij}}{\sqrt{\alpha + d_i} \sqrt{\alpha + d_j}} - \frac{A_{ij}}{\sqrt{d_i d_j}}, & i \neq j \\ \frac{\alpha}{\alpha + d_i}, & i = j \end{cases}
\tag{34}
$$

Following the notion of the Theorem 4 stated in (Karhadkar et al., 2022) we can assume $x$ is a normalized eigenvector of $A_N$ with corresponding eigenvalue $\lambda$. Therefore, the first-order change in the corresponding eigenvalue can be represented as:

$$
\begin{aligned}
x^\top (\delta A_N) x &= \sum_{i \neq j} \delta A_N[i][j] x_i x_j + \sum_{i=j} \delta A_N[i][j] x_i^2 \\
&= \sum_{i \neq j} \left( \frac{A_{ij}}{\sqrt{\alpha + d_i} \sqrt{\alpha + d_j}} - \frac{A_{ij}}{\sqrt{d_i d_j}} \right) x_i x_j \\
&\quad + \sum_{i=j} \frac{\alpha}{\alpha + d_i} x_i^2 \\
&= \sum_{i \neq j} \left( \frac{1}{\sqrt{\alpha + d_i} \sqrt{\alpha + d_j}} - \frac{1}{\sqrt{d_i d_j}} \right) A_{ij} x_i x_j \\
&\quad + \sum_{i=j} \frac{\alpha}{\alpha + d_i} x_i^2
\end{aligned}
\tag{35}
$$

Consider,

$$
\begin{aligned}
F_{ij}^{(1)}(\alpha) &= \left( \frac{1}{\sqrt{\alpha + d_i}\sqrt{\alpha + d_j}} - \frac{1}{\sqrt{d_i d_j}} \right) \\
F_i^{(2)}(\alpha) &= \frac{\alpha}{\alpha + d_i} x_i^2
\end{aligned}
\tag{36}
$$

If $d_i, d_j \gg \alpha$, then $F_{ij}^{(1)}(\alpha) \approx 0$ which lead to

$$
x^\top (\delta A_N) x \approx F_i^{(2)}(\alpha) \approx \sum_{i=j} \frac{\alpha}{\alpha + d_i} x_i^2.
\tag{37}
$$

If $\alpha$ increases then $F_i^{(2)}(\alpha)$ also increases. This reflects the change in the eigenvalue of $A_N^\alpha$ increases, indicating the decrease in the eigenvalues of the $\tilde{L}_\alpha$. $\qquad\square$

**Theorem 4** Consider a connected graph $\mathcal{G}$ with normalized adjacency matrix $A_N = D^{-\frac{1}{2}} A D^{-\frac{1}{2}}$. Assuming each element of $A$ except the diagonal is multiplied by $1 + \gamma$ where $\gamma > 0$ is a significantly small quantity, then the updated normalized adjacency matrix will be $A_N^\gamma$. The change in the eigenvalues of $A_N^\gamma$ with respect to the eigenvalues of $A_N$ will decrease when $\gamma$ increases.

*Proof.* If the non-diagonal elements of $A$ are multiplied by $1 + \gamma$, then the symmetrically normalized graph Laplacian will be,

$$
\tilde{L}_\gamma = \mathrm{I} - \tilde{D}_\gamma^{-\frac{1}{2}} \tilde{A}_\gamma \tilde{D}_\gamma^{-\frac{1}{2}},
\tag{38}
$$

where $\tilde{A}_\gamma = (\gamma + 1)A + \mathrm{I}$ and $\tilde{D}_\gamma = (\gamma + 1)D + \mathrm{I}$. Let us denote the normalized adjacency matrix without self-loops as $A_N = D^{-\frac{1}{2}} A D^{-\frac{1}{2}}$. The element of $A_N$ is represented as:

$$
A_N[i][j] = \begin{cases} \frac{A_{ij}}{\sqrt{d_i d_j}}, & i \neq j \\ 0, & i = j \end{cases}
\tag{39}
$$

After multiplying $(1 + \gamma)$ to the non-diagonal elements of $A$ and adding one self-loops, the normalized adjacency will be $A_N^\gamma = \tilde{D}_\gamma^{-\frac{1}{2}} \tilde{A}_\gamma \tilde{D}_\gamma^{-\frac{1}{2}}$. The elements of $\tilde{A}_N^\gamma$ can be represented as:

$$
A_N^\gamma[i][j] = \begin{cases} \frac{(\gamma+1)A_{ij}}{\sqrt{1+(\gamma+1)d_i}\sqrt{1+(\gamma+1)d_j}} & i \neq j \\ \frac{1}{1+(1+\gamma)d_i} & i = j \end{cases}
\tag{40}
$$

The entry-wise change in the normalized adjacency matrix is presented as:

$$
\delta A_N[i][j] = \begin{cases} \frac{(\gamma+1)A_{ij}}{\sqrt{1+(\gamma+1)d_i}\sqrt{1+(\gamma+1)d_j}} - \frac{A_{ij}}{\sqrt{d_i d_j}}, & i \neq j \\ \frac{1}{1+(1+\gamma)d_i}, & i = j \end{cases}
\tag{41}
$$

Following the notion of the Theorem 4 stated in (Karhadkar et al., 2022) we can assume $x$ is a normalized eigenvector of $A_N$ with corresponding eigenvalue $\lambda$. Therefore, the first-order change in the spectral gap can

be represented as:

$$
\begin{aligned}
x^\top(\delta A_N)x &= \sum_{i \neq j} \delta A_N[i][j]x_i x_j + \sum_{i=j} \delta A_N[i][j]x_i^2 \\
&= \sum_{i \neq j} \left( \frac{(\gamma+1)A_{ij}}{\sqrt{1+(\gamma+1)d_i}\sqrt{1+(\gamma+1)d_j}} - \frac{A_{ij}}{\sqrt{d_i d_j}} \right) x_i x_j \\
&\quad + \sum_{i=j} \frac{1}{1+(1+\gamma)d_i}x_i^2 \\
&= \sum_{i \neq j} \left( \frac{(\gamma+1)}{\sqrt{1+(\gamma+1)d_i}\sqrt{1+(\gamma+1)d_j}} - \frac{1}{\sqrt{d_i d_j}} \right) A_{ij} x_i x_j \\
&\quad + \sum_{i=j} \frac{1}{1+(1+\gamma)d_i}x_i^2
\end{aligned}
\tag{42}
$$

Consider the following,

$$
\begin{aligned}
F_{ij}^{(1)}(\gamma) &= \left( \frac{(\gamma+1)}{\sqrt{1+(\gamma+1)d_i}\sqrt{1+(\gamma+1)d_j}} - \frac{1}{\sqrt{d_i d_j}} \right) \\
F_{i}^{(2)}(\gamma) &= \frac{1}{1+(1+\gamma)d_i}x_i^2
\end{aligned}
\tag{43}
$$

Now we can rewrite,

$$
\begin{aligned}
F_{ij}^{(1)}(\gamma) &= \left( \frac{(\gamma+1)}{\sqrt{1+(\gamma+1)d_i}\sqrt{1+(\gamma+1)d_j}} - \frac{1}{\sqrt{d_i d_j}} \right) \\
&= \left( \frac{1}{\sqrt{\frac{1}{1+\gamma}+d_i}\sqrt{\frac{1}{1+\gamma}+d_j}} - \frac{1}{\sqrt{d_i d_j}} \right)
\end{aligned}
\tag{44}
$$

As we mentioned $\gamma$ is a sufficiently small quantity and with $d_i, d_j \gg 1$, then $F_{ij}^{(1)}(\gamma) \approx 0$. Now we have,

$$
x^\top(\delta A_N)x \approx F_i^{(2)}(\gamma) \approx \frac{1}{1+(1+\gamma)d_i}x_i^2
\tag{45}
$$

If $\gamma$ increases then $F_i^{(2)}(\gamma)$ decreases reflecting the change in the eigenvalues of $\tilde{A}_N^\gamma$ decreases. This indicates the increase in the eigenvalues of $\tilde{L}_N^\gamma$. $\qquad\square$

