# OpenReview forum: "Learning from Heterophilic Graphs: A Spectral Theory Perspective on the Impact of Self-Loops and Parallel Edges"
_TMLR — Rejected by TMLR_

### Review · Reviewer_zGLg · 2025-05-12

**Summary Of Contributions:**

This paper studies the performance trends of Graph Convolutional Networks (GCNs) on heterophilic graphs, considering the impact of self-loops and parallel edges. By categorizing graphs based on whether GCN performance improves or deteriorates, the authors gain insights into the spectra and unique structures of these graphs. They assert that this approach offers an alternative method for evaluating graph spectra and properties, bypassing the need for eigenvalue decomposition.

**Audience:**

Yes

**Claims And Evidence:**

Yes

**Requested Changes:**

I suggest the following changes:
- The authors should provide a more rigorous discussion on why different graphs exhibit varying GCN performance trends. In particular:

  - What characteristics or properties of the input graphs are reflected in these performance trends?
  - Why do different graphs produce different trends, and why do these differ from those observed in random graphs? Under what conditions do the eigenvalues and GCN performance increase or decrease as more self-loops or parallel edges are added?
  - What new insights do these trends offer that cannot be efficiently obtained using traditional algorithms?

- Additionally, the authors should include a runtime comparison between their approach and standard algorithms like eigenvalue decomposition to better demonstrate the efficiency of their method.

- Minor typos:

  - In Lemma 1, $\alpha_1$ should be $\alpha$.
  - In Lemma 2, "self-loops" should be corrected to "parallel edges".

**Strengths And Weaknesses:**

Strengths:

- The authors provide theoretical foundations on the impact of self-loops and parallel edges on the graph spectrum, verifying their results on random graphs.
- They conduct experiments on a diverse range of heterophilic graph datasets, offering a comprehensive analysis of GCN performance trends and their relationship to the graph spectrum and properties.

Weaknesses:

- The authors claim that:
  > The categorization of performance trends leads to the identification of characteristics of the graph spectrum. Different performance trends underscore the various patterns of the spectrum which reveals the properties of the networks like connected components, community structure, sparsity, etc.

  However, it’s unclear why evaluating GCNs on graphs with varying self-loops and parallel edges is necessary to determine these characteristics. Traditional algorithms can efficiently compute these properties, and I don’t see the need for multiple GCN training sessions to categorize input graphs.

  Furthermore, the discussion on the connection between GCN performance and graph spatial properties appears largely empirical. The authors analyze GCN performance trends on various heterophilic graph datasets and observe some discrepancies from random graphs. However, they lack sufficient explanations for these differences and theoretical insights into how these trends relate to spectral and spatial graph properties. While they present case studies illustrating commonalities across different categories, these discussions lack rigor and provide limited insights.
- The theoretical part of this paper focuses solely on regular graphs. For general graphs, the authors only provide upper bounds for the maximum eigenvalues and results for significantly small perturbations. In Remark 1, they claim that:
  > The maximum eigenvalue of $\tilde{L}\_\alpha$ decreases with the increase in self-loops, indicating a shrinking of the graph spectrum. Conversely, the maximum eigenvalue of $\tilde{L}_\gamma$ increases with the increase in parallel edges, suggesting an expansion of the graph spectrum.

  However, this statement lacks rigor since the authors only demonstrate the increase or decrease of the upper bound of the maximum eigenvalue.

  Furthermore, the performance trends of GCNs on various real-world graphs differ from those of random graphs, contradicting the authors’ theories. The paper lacks theoretical discussion on the connection between graph properties and the impact of adding self-loops and parallel edges on the spectrum and GCN performance. Given that the effect of adding edges on the graph spectrum has been extensively studied in prior works [1], I expect more discussions on this topic.
- The author asserts that their approach is significantly more efficient than eigenvalue decomposition:

  > By separately observing the performance trends of low-pass filters with the addition of self-loops and parallel edges, it is possible to explore the eigenvalue distribution. This approach drastically reduces the computational burden and offers deeper insights into the intricate structural patterns of the given networks.

  However, this claim lacks empirical evidence. Eigenvalue decomposition has polynomial complexity, and there are algorithms that compute the top $k$ eigenvalues, which is more efficient than computing the entire decomposition. The authors should provide evidence demonstrating that their approach “drastically reduces the computational budget” of traditional algorithms.

---
**References**

[1] Guo, J. M., Tong, P. P., Li, J., Shiu, W. C., & Wang, Z. W. (2018). The effect on eigenvalues of connected graphs by adding edges. *Linear Algebra and its Applications*, 548, 57-65.

---

> ### Author Response · Authors · 2025-05-27
> **R1: Response to Reviewer zGLg for Paper4634**
>
> We appreciate the reviewer's insightful comments and pertinent feedback regarding our work. We provide the point-by-point response to the queries as follows,
>
> __Query__: However, this statement lacks rigor since the authors only demonstrate the increase or decrease of the upper bound of the maximum eigenvalue.
>
> **Our response**:
> > We appreciate the concern of the reviewer regarding the theoretical analyses in our work. Our analyses showed the increase and decrease in the upper bound of the maximum eigenvalue when parallel edges and self-loops are added respectively, not showing the exact changes in the eigenvalues. We also empirically found that similar effects on the maximum eigenvalues for the multiple random graphs where every eigenvalue increases or decreases as per the edge addition. Yet, our analysis does not include any insights on the exact changes in every eigenvalue in the case of general graphs. To address this gap, we offer detailed proofs for the regular graphs where we can prove the desired effects on each eigenvalue for the addition of self-loops and parallel edges. Furthermore, as supporting findings, we showed the changes in the eigenvalues for the small perturbation for general graphs. We believe the proof for the general graphs can be posed as a future research problem in the landscape of graph learning.
> > We experimented on Chameleon with perturbation $\alpha=0.0001$ and $\gamma=0.0001$, and we observed similar trends as observed in your work.
> | Dataset | $A +\alpha I$ | $A + 2\alpha I$ | $A + 3\alpha I$ | $A+4 \alpha I$  | $A+5\alpha I$ | Trend |
> |------------|---------------|---------------|---------------|---------------|---------------|------------|
> |        | 64.64 ± 1.65  | 63.15 ± 1.76  | 61.59 ± 2.02  | 59.91 ± 1.74  | 58.50 ± 1.94  | Decreasing ($\downarrow$) |
> |Chameleon| $A +\alpha I $    |  $2 \gamma A + I $ | $3 \gamma A + I $ | $4 \gamma A + I $ | $5 \gamma A + I $
> |              | 47.61 ± 2.27  | 47.82 ± 2.33  | 47.94 ± 1.94  | 48.20 ± 2.12  | 48.20 ± 2.49  | Increasing ($\uparrow$) |
>
> __Query__: Furthermore, the discussion on the connection between GCN performance and graph spatial properties appears largely empirical. The authors analyze GCN performance trends on various heterophilic graph datasets and observe some discrepancies from random graphs. However, they lack sufficient explanations for these differences and theoretical insights into how these trends relate to spectral and spatial graph properties. While they present case studies illustrating commonalities across different categories, these discussions lack rigor and provide limited insights.
>
> **Our response**:
> > We appreciate the valuable feedback of the reviewer. The sole objective of our work is to offer insights into the graph spectra without performing expensive eigendecomposition. The performance trends can mostly be attributed to the gradual addition of self-loops or parallel edges. These edge additions shift the graph spectrum toward either the minimum value zero or the maximum value $2$. The shifting of spectra alters the parity between the lower and higher frequencies. The altered parity affects the performance of low-pass filters like GCN. The increasing  The gradual addition of self-loops or parallel edges paves the way for either increasing or decreasing lower or higher frequencies. This change in parity is instrumental in deciding the performance trends of the underlying heterophilic graphs. We first map initial graph spectra with four different performance trends by experimenting on small and medium-sized graphs. Later, we further map the graph spectra based on the performance trends of GCN on large-scale graphs.
>
> __Query__: Traditional algorithms can efficiently compute these properties, and I don’t see the need for multiple GCN training sessions to categorize input graphs
>
> **Our response**:
> > We appreciate the concern of the reviewer regarding the necessity of training multiple GCN sessions to identify graph properties. Our primary objective is crafting a pathway for understanding the distribution of eigenvalues in graph spectra. To address the problem, we pursue multiple GNN sessions by incorporating self-loops and parallel edges in the underlying networks. The performance trends that are evaluated in the experiments are pivotal in offering deeper insights into the several properties of the graph spectrum like the shape of the spectrum, skewness of the distribution, parity of the eigenvalues, etc. Understanding these properties also delineates the other secondary properties like community structure, connected components, average degree, isolated nodes, etc. We have a consensus with the reviewer that this set of properties can easily be estimated by traditional graph algorithms. Yet, our task is not to discover these properties, but rather to unfold the characteristics of spectra. These properties can be regarded as the "by-product" of understanding graph spectra which is our sole target.

---

> ### Author Response · Authors · 2025-05-27
> **R1: Response to Reviewer zGLg for Paper4634: PART 2**
>
> __Query__: Why do different graphs produce different trends, and why do these differ from those observed in random graphs? Under what conditions do the eigenvalues and GCN performance increase or decrease as more self-loops or parallel edges are added?
>
> **Our response**:
> > The initial distribution of eigenvalues is pivotal for the performance trends observed in the performance of the GCN. The addition of self-loops or parallel edges in the graph structure that leads to a distribution shift in the spectrum. Thus, GCN exhibits different patterns in performance according to the modified graph spectrum.
>
> > In this work, we solved node classification tasks in the ambit of semi-supervised or supervised settings. The class labels for the nodes of the graphs maintain a strong correlation with the node features and these datasets are also leading benchmarks. Therefore, patterns observed in our experiments are comprehensive and significant. Though similar experimentation on random graphs may not yield such performance trends. This happens due to the absence of feature to label correlation, unlike the prevailing benchmarks. Notably, the effects of adding self-loops and parallel edges on the graph spectrum are universally valid for all categories of graphs, irrespective of having node labels.
>
> __Query__: Why different graphs exhibit varying GCN performance trends?
>
> **Our response**:
> > GCN enhances the lower eigenvalues of the symmetrically normalized graph Laplacian, acting as a low-pass filter. The node features of the homophilic graphs are smoothed out while the low-pass filter (here GCN) is applied, forming a well-defined cluster structure. On the contrary, node features of the heterophilic graphs are sharpened when the high-pass filter is applied, creating deterrents to form clusters. We also observe that the gradual addition of self-loops or parallel edges into the graph will alter the spectrum of the network. This will also impact the parity of lower and higher frequencies and alter the distribution of the graph spectrum. When GCN is applied to this modified graph, the GCN amplifies the lower frequencies, and if the updated spectrum contains more lower frequencies, the performance will boost. The performance trends depend only on the initial distribution of the graph spectrum. Therefore, different graphs demonstrated various performance trends.
>
> __Query__: What new insights do these trends offer that cannot be efficiently obtained using traditional algorithms?
>
> **Our response**:
> > We map the input graph into one of the four categories based on the patterns observed in the performance of GCN. In this work, we primarily focused on determining the eigenvalue distribution of the graph spectrum. Pursuing costly eigenvalue decomposition may raise computational constraints, especially for large graphs. These limitations are also evident in our runtime comparison experiments. We summarize the following advantages we can have without performing direct eigendecomposition as,
> > 1. Our strategy reveals the shape of the graph spectrum characterizing structural properties like the presence of connected components, communities, etc. For example, regular graphs have a symmetric graph spectrum. Additionally, the spectral density can identify graph classes like scale-free, random, and small-world.
> 2. Computing similarity measures between two large graphs is cumbersome. Our method offers insights into the spectrum distributions, whose comparisons enable quick similarity assessment between the large networks.
> 3. We can detect anomalies of the evolving or dynamic networks by monitoring shifts in the respective distribution shifts.

---

> ### Author Response · Authors · 2025-05-27
> **R1: Response to Reviewer zGLg for Paper4634: PART 3**
>
> __Query__: Additionally, the authors should include a runtime comparison between their approach and standard algorithms like eigenvalue decomposition to better demonstrate the efficiency of their method.
>
> **Our response**: We thank the reviewer for suggesting adding a comparative study on the runtime between traditional eigendecomposition algorithms and our method. We offered a vivid demonstration in the following way
>
> > ### Runtime Comparison with Traditional Eigendecomposition Algorithms
> We conducted an extensive study on $17$ heterophilic benchmarks to compare the runtime of our proposed strategy with that of traditional eigendecomposition algorithms. We applied an inbuilt function \texttt{np.linalg.eig} implemented in the Numpy package to execute eigendecomposition. All experiments are carried out on a single $24$ NVIDIA GeForce RTX 3090 GPU and we obtained the results as presented in Table 8. For each graph, we leverage $10$ GNN training sessions for $5$ self-loops and $5$ parallel edges. For both cases, we utilized \texttt{time} function from Python. The analyses reveal that for small-scale graphs, the traditional algorithm outperforms our approach but for medium-sized graphs, our approach exhibited faster runtime in comparison to the inbuilt algorithm. Additionally, for large-scale graphs, our system showed Out-of-Memory (OOM) or process killed and was unable to estimate eigendecomposition. The experimental results underscore the significance of our approach to gain insights into the spectra without performing the time and memory-intensive existing eigendecomposition techniques.
>
> ### 📊 Comparative Runtime Study (Part 1)
>
> | Runtime (sec.)       | Chameleon | Squirrel | Film     | Texas   | Cornell | Wisconsin |
> |----------------------|-----------|----------|----------|---------|---------|-----------|
> | `np.linalg.eig`      | 2.5334    | 22.1533  | 86.9033  | 0.0442  | 0.0226  | 0.0599    |
> | **Ours**             | 73.4782   | 162.0774 | 371.0116 | 52.9237 | 52.5499 | 52.9963   |
>
> ### 📊 Comparative Runtime Study (Part 2)
>
> | Runtime (sec.)       | arxiv-year | snap-patents | Penn94    | pokec     | twitch-gamers | genius   |
> |----------------------|------------|--------------|-----------|-----------|----------------|----------|
> | `np.linalg.eig`      | OOM        | OOM          | OOM       | OOM       | OOM            | OOM      |
> | **Ours**             | 365.9806   | 699.1057     | 2866.6932 | 1467.4539 | 1097.2305      | 576.4411 |
>
> ### 📊 Comparative Runtime Study (Part 3)
>
> | Runtime (sec.)       | Roman-empire | Amazon-ratings | Minesweeper | Tolokers  | Questions     |
> |----------------------|--------------|----------------|-------------|-----------|----------------|
> | `np.linalg.eig`      | 3240.4251    | 3032.0099      | 40.7877     | 94.4341   | process killed |
> | **Ours**             | 95.5101      | 135.3722       | 79.7196     | 360.0264  | 247.7609       |
>
> __Query__: Given that the effect of adding edges on the graph spectrum has been extensively studied in prior works [1], I expect more discussions on this topic.
>
> **Our response**: We thank the reviewer for suggesting the prior work [The effect on eigenvalues of connected graphs by adding edges. Linear Algebra and its Applications, 548, 57-65.] and we observe the following notable differences with our work,
>
> A. The paper deals with the eigenvalues of the adjacency matrix and makes no mention of the normalized Laplacian or graph spectrum. In contrast, our whole work revolves around determining the spectrum properties and the eigenvalues of the symmetrically normalized graph Laplacian.
>
> B. The work studied the effects of the addition of any edges in the graph on the eigenvalues of the adjacency matrix. No further research is conducted on the effect of self-loops and parallel edges. In our work, we systematically present the utility of adding self-loops and parallel edges to discover patterns in the heterophilic networks.
>
> C. The prior work does not offer any deduction regarding the distribution of the graph spectrum in the ambit of heterophilic graphs. Our work explores the connection between graph spectra ns the performance of low-pass filters (like GCN).
>
> We discuss these further in the Related Work section and cite the paper in our updated manuscript. The aforementioned gaps justify contextualizing our work in the graph learning paradigm.
>
> **Please note that all the responses mentioned in this rebuttal are updated in the revised manuscript marked in blue colored text.**

---

### Review · Reviewer_GPRq · 2025-05-13

**Summary Of Contributions:**

This paper explores the behavior of message-passing graph neural networks, specifically graph convolutional networks (GCN), on heterophilic graphs through a spectral lens. The authors introduce a novel analytical framework for studying how modifications to graph topology—via self-loops and parallel edges—affect the spectral properties of the graph Laplacian. Then, they study the performance of low-pass filters like GCN. They categorize performance trends into four distinct types based on whether self-loops or parallel edges are added, and relate these trends to eigenvalue distributions. Empirical studies on 17 benchmark heterophilic datasets and accompanying theoretical derivations strengthen their conclusions. The work offers a cost-effective strategy for estimating spectral properties without computing full eigendecompositions.

**Audience:**

Yes

**Broader Impact Concerns:**

None. The paper focuses on theoretical analysis and algorithmic evaluation without raising ethical or resource concerns.

**Claims And Evidence:**

Yes

**Requested Changes:**

* Please clarify in Section 3.1 the distinction between $\widetilde{A}$, $A_\gamma$, and $A^\alpha$; their current presentation assumes too much familiarity.
* The derivation in Lemma 1 references Proposition 1 but requires more explicit connection between assumptions and bounds.
* In Section 4.3, consider reporting raw average degree and edge density in addition to the log-scaled versions for easier cross-comparison with prior literature.
* Table 1 could benefit from a clearer explanation of how these categories are assigned—perhaps add a short example.
* Section 4.10 (variation of both self-loops and parallel edges) is an interesting empirical contribution and might deserve more theoretical insight or discussion.
* Please proofread for small typographical errors: e.g., "noticessly" → "notably", and "spG" sometimes appears as both a variable and a label.
* Minor issues: the plots in Figures 2–5 would be more accessible to colorblind readers if color palettes with greater luminance contrast were used.

**Strengths And Weaknesses:**

Strengths:
- The paper provides a compelling connection between spectral graph theory and practical GCN performance on heterophilic graphs, which is an underexplored area.
- Paper is well-written and the theoretical derivations are well-motivated and complemented by empirical results.
- The use of multiple large-scale and standard benchmark datasets gives the experimental validation strong breadth.
- The performance categorization (A, B, C, D) is a useful abstraction that could influence future GNN design or hyperparameter tuning practices.


Weaknesses:
- The notation is occasionally inconsistently defined across sections (e.g., varying notations for Laplacian variants and adjacency matrices with/without loops).

- The theoretical exposition, while rigorous, could benefit from improved structure—some lemmas are buried deeply in dense algebra math.
-  I suggest combining subplots or using clearer legends may improve interpretability for some figures (e.g., Figure 4 and 5)

---

> ### Author Response · Authors · 2025-05-27
> **R1: Response to Reviewer GPRq for Paper4634**
>
> We appreciate the reviewer's insightful comments and valuable feedback regarding our work. We provide the point-by-point response to the queries as follows,
>
> __Query__:  Please clarify in Section 3.1 the distinction between $\widetilde{A}$, $A_\gamma$, and $A^\alpha$; their current presentation assumes too much familiarity.
>
> **Our response**: We thank the reviewer for identifying the inconsistency in the notations. We add a table explaining every notation used in this work to resolve the notational ambiguity. Please refer to Table 2 for a better understanding of our work.
>
> __Query__: The derivation in Lemma 1 references Proposition 1 but requires more explicit connection between assumptions and bounds.
>
> **Our response**: We thank the reviewer for the suggestion. We have some notational flaws in Proposition 1 which may hinder comprehension of the connection between Lemma 1 and Proposition 1. We identified and resolved the fallacies. Furthermore, we have identified a similar issue in Lemma 2. We also fixed those errors. The modifications can be found in the Appendix section.
>
> __Query__: In Section 4.3, consider reporting raw average degree and edge density in addition to the log-scaled versions for easier cross-comparison with prior literature.
>
> **Our response**: We mentioned the original average node degree and edge density for all $17$ heterophilic graphs. The updated data can be found in Table 3 marked with blue color.
>
> __Query__: Table 1 could benefit from a clearer explanation of how these categories are assigned—perhaps add a short example.
>
> **Our response**: We appreciate the suggestion of the reviewer for explaining the assigning of performance trends with an illustrative example. We mentioned it as follows,
>
> > We aim to bridge the gap by offering two simple strategies that update the graph topology by incorporating self-loops and parallel edges. After the alteration of edge connections, Graph Convolution Network (GCN), a recognized and well-adopted low-pass filter, is applied to the updated graph. We observe some interesting patterns in the performance trends of GCN when the number of self-loops or parallel edges gradually increases. The performance either monotonically improves or degrades by adding either self-loops or parallel edges. Therefore, we can have four distinct combinations of performance trends. Each combination is tagged with a category name, which is mentioned in the Table 1. The category assignment is purely dependent on the combination of performance trends in both self-loops and parallel edge addition. For instance, if GCN shows an increasing trend on self-loop addition and a decreasing trend on parallel edge addition, then the underlying dataset is categorized as "B". Furthermore, for parallel edge addition, if the trend changes to increasing, then the category will change to "A". Importantly, every category delineates a particular set of characteristics regarding the spectrum of the input graph.
>
> __Query__: Section 4.10 (variation of both self-loops and parallel edges) is an interesting empirical contribution and might deserve more theoretical insight or discussion.
>
> **Our response**: We appreciate the reviewer's feedback. In this context, we offer an intuitive explanation behind the performance trends of GCN with varying both self-loops and parallel edges.
>
> > ### Intuitive Explanation
> Adding self-loops increases the number of lower frequencies, shifting the graph spectrum towards the eigenvalue $0$. Conversely, the addition of parallel edges increases the number of higher frequencies, shifting the spectrum toward the value $2$. The addition of a maximum $\mathcal{P}$ number of self-loops and a maximum $\mathcal{Q}$ number of parallel edges shifts the graph spectrum to the left and right directions accordingly. Precisely, the addition of $\alpha \le \mathcal{P}$-number of self-loops and $\gamma \le \mathcal{Q}$-number of parallel edges will transform the graph spectrum intermediate between the two aforementioned extreme scenarios. The updated spectrum oscillation between the two extreme cases and the performance of GCN is also reflected in our quantitative analysis.
>
> __Query__: Please proofread for small typographical errors: e.g., "noticessly" → "notably", and "spG" sometimes appears as both a variable and a label.
>
> **Our response**: We rectified the typographical errors and cross-checked multiple times to prevent disruption of future readers.
>
> __Query__:  Minor issues: the plots in Figures 2–5 would be more accessible to colorblind readers if color palettes with greater luminance contrast were used.
>
> **Our response**: We appreciate the reviewer's feedback to improve the color contrast of some of the figures. We have used $6$ different colors for six datasets and updated the Figures accordingly in the manuscript.
>
>
> **Please note that all the responses mentioned in this rebuttal are updated in the revised manuscript marked in blue colored text.**

---

### Review · Reviewer_HbQ3 · 2025-05-13

**Summary Of Contributions:**

This work studies the spectral behavior of graph convolutional networks (GCNs) on heterophilic graphs. The authors analyze how GCN performance changes when graph topology is modified by adding self-loops and parallel edges. These modifications are shown, both theoretically and empirically, to induce predictable shifts in the graph Laplacian spectrum. Since GCNs act as low-pass filters, these spectral shifts can affect performance. They validate their theoretical predictions with experiments on 17 datasets and categorize four performance trends ($\uparrow\uparrow$, $\uparrow\downarrow$, $\downarrow\uparrow$, $\downarrow\downarrow$).

**Audience:**

Yes

**Claims And Evidence:**

Yes

**Requested Changes:**

* **Section 3.6 (important)**: The theoretical results in Section 3.6 could benefit from a clearer narrative. We suggest the authors consider:
  * Introducing each result with an intuitive summary (e.g., "this shows adding self-loops shifts the spectrum like so").
  * Adding a few remarks on why these spectral shifts might affect performance in the context of GCNs as low-pass filters.
  * Sketching how these results relate to the empirical trends observed in the paper (e.g., "our theoretical results predict that... this is confirmed in datasets like...").
* **Provide actionable suggestions to practitioners**: It would be helpful to provide some discussion on what the four trend categories (A–D) suggest in practice. Even speculative guidelines would be good.

**Strengths And Weaknesses:**

## Strengths

The work is well-motivated, theoretically sound, and of interest to the graph learning community.  It is a nice contribution to the theoretical understanding of GCNs. The empirical results validate the theoretical analysis.

## Weaknesses

* **Section 3.6 presentation (----)**: The theoretical results are presented without sufficient discussion. Key insights on the spectrum shifts are only discussed later. The connection to the empirical results is not clearly laid out for the reader.
* **Lack of actionable outcome for practitioners (-)**: The authors do not spell out guidelines on how the A-D categories could inform graph preprocessing and so on.

---

> ### Author Response · Authors · 2025-05-27
> **R1: Response to Reviewer HbQ3 for Paper4634**
>
> We thank the reviewer for providing pertinent remarks for our work. We provide the point-by-point response for the queries as follows,
>
> __Query__: Section 3.6 (important): The theoretical results in Section 3.6 could benefit from a clearer narrative. We suggest the authors consider: Introducing each result with an intuitive summary (e.g., "this shows adding self-loops shifts the spectrum like so").
>
> **Our response**: We appreciate the reviewer's concern and elaborate the implications of theorems in the form of remarks.
>
> > ### Remark 3:
> We have shown that for the regular graphs, the eigenvalues of the symmetrically normalized graph Laplacian decrease concurrently when self-loops are added. Adding self-loops attenuates the graph spectrum frequencies, thereby shifting the graph spectrum towards zero eigenvalue. This will also increase the number of lower frequencies and decrease the number of higher frequencies.  The eigenvalues of the symmetrically normalized graph Laplacian increase when parallel edges are added. Adding parallel edges amplifies the frequencies of the graph spectrum, which shifts the spectrum toward the value $2$. Consequently, the number of lower frequencies decreases and the number of higher frequencies increases.
>
> > ### Remark 4:
>  Theorem 3 suggests that the change in the eigenvalues of the normalized adjacency matrix increases an increasing number of self-loops which also signifies the decrease in the change in the eigenvalues of $\tilde{L}$ . The small perturbation in the adjacency matrix leads to a shrinking of the frequencies in the graph spectrum. The spectrum will shift toward the zero eigenvalue. Spectrum shrinking also highlights the alteration in the parity of frequencies. The number of lower frequencies increases and the number of higher frequencies decreases. Conversely, Theorem 4 states that the change of eigenvalues of the normalized adjacency matrix decreases with the increasing number of parallel edges. This indicates an increase in the change in the eigenvalues of $\tilde{L}_{\gamma}$, highlighting the spectrum expansion. To this effect, the graph spectrum will shift toward the value $2$. Furthermore, the parity of frequencies changes, specifically, the number of lower frequencies decreases and the number of higher frequencies increases.
>
> __Query__: Adding a few remarks on why these spectral shifts might affect performance in the context of GCNs as low-pass filters.
>
> **Our response**: The spectral shifts alters the parity of lower and higher eigenvalues of the normalized Laplacian, which impacts the performance of GCN. We add a separate section to discuss the effects of spectral shifts on the performance of GCN.
> > ### Effect on the Performance of GCN for Spectral Shifts
> The theoretical analyses offer insights into the effects on graph spectra when self-loops or parallel edges are added. We also observed that spectrum shifts either to zero eigenvalue or to the value $2$ and also alter the parity of the frequencies or eigenvalues. The higher number of low frequencies in a network indicates the presence of well-separated communities, mostly sharing identical node labels. GCN or any low-pass filter is supposed to smooth out the features of the connected nodes, assuming that they share similar class labels. The addition of self-loops increases the number of lower frequencies which appears to be beneficial for smoothing out features, improving the performance of GCN. Conversely, the graph spectra shift toward the value $2$, signifying the increase in the higher frequencies. A network with overlapped communities contains many high frequencies in the spectrum. GCN is poised to smooth out the features of nodes that are probably having different class labels. This will lead to the degradation of the performance of GCN.
>
> **Please note that all the responses mentioned in this rebuttal are updated in the revised manuscript marked in blue colored text.**

---

> ### Author Response · Authors · 2025-05-27
> **R1: Response to Reviewer HbQ3 for Paper4634 : PART 2**
>
> Query: Sketching how these results relate to the empirical trends observed in the paper (e.g., "our theoretical results predict that... this is confirmed in datasets like...").
>
> **Our response**: We add a section to illustrate the connection between theoretical implications and the evaluated performance of GCN on the heterophilic graphs.
> > ### Connection between Theoretical Analysis and Performance Trends
> The various performance trends depend on the presence of parity between lower and higher frequencies in the spectrum of the input graph. As discussed in the theoretical analyses, the addition of self-loops and parallel edges respectively decreases and increases the eigenvalues or frequencies in the spectrum. If GCN witnessed an increasing trend on a heterophilic graph with the gradual addition of self-loops, then we can conclude that initially the graph spectrum is aligned toward zero eigenvalue and the addition of self-loops further tilts the balance to lower frequencies. The similar effects are evident in datasets like Actor, Cornell, pokec, etc. In contrast, if the performance of GCN exacerbates with the gradual addition of parallel edges, then we can infer initial graph spectrum is skewed towards the maximum value $2$, indicating a greater number of higher frequencies. Further addition of parallel edges enhances the high frequencies and the decreasing trend observed in the performance of GCN. The impact is evident in datasets like Actor, arxiv-year, snap-patents, Tolokers, etc.
>
> __Query__: It would be helpful to provide some discussion on what the four trend categories (A–D) suggest in practice. Even speculative guidelines would be good.
>
> **Our response**: We thank the reviewer for suggesting to discuss the advantages of our work from the practitioners' perspective. We discuss a few aspects that may benefit to understand spectral properties of the large-scale graphs. We also found our approach can detect anomalies for dynamic graphs. We add a separate section to illustrate the benefits.
>
> > In this work, we primarily focus on determining properties of the graph spectrum without resorting to costly eigenvalue decomposition. We map the input graph into one of the four categories based on the patterns observed in the performance of GCN. Beyond the theoretical insights and empirical evaluations, our work possesses some benefits for practitioners. We enjoy the following advantages without performing direct eigendecomposition.
> > 1. Our strategy reveals the shape of the graph spectrum characterizing structural properties like the presence of connected components, communities, etc. For example, regular graphs have a symmetric graph spectrum. Additionally, the spectral density can identify graph classes like scale-free, random, and small-world.
> 2. Computing similarity measures between two large graphs is cumbersome. Our method offers insights into the spectrum distributions, whose comparisons enable quick similarity assessment between the large networks.
> 3. We can detect anomalies of the evolving or dynamic networks by monitoring shifts in the respective distribution shifts.
>
> **Please note that all the responses mentioned in this rebuttal are updated in the revised manuscript.**

---

### Decision · Action_Editor_DN65 · 2025-06-09

**Recommendation:** Reject

**Additional Comments:**

If the authors choose to resubmit, I strongly urge them to consider a precise narrative in the paper, to be highlighted in the introduction, and to avoid vague or fuzzy statements (e.g., on the performance of GCNs) which are not fully validated by the experiments.

**Audience:**

Yes

**Audience Explanation:**

The paper is potentially appealing to both the GNN community and researchers interested in spectral graph theory. However, in its current form the paper does not provide a cohesive narrative.

**Claims And Evidence:**

No

**Claims Explanation:**

The paper deals with the relation between (a) the spectra of heterophilic graphs; (b) their variation when adding self-loops or "parallel edges"; (c) the performance of graph convolutional networks (GCNs). Multiple parts of the paper deal with these three topics separately, including (1) an analysis of (b) on random graphs; (2) several lemmas on the upper bounds of eigenvalues on regular graphs; (3) a method to evaluate (a) based on the performance of a GCN when evaluated on (b).

We received three reviews. Two of them (HbQ3, GPRq) had mostly notational or narrative issues, that were solved in the rebuttal. The third reviewer (zGLg) had more serious concerns that were not resolved during rebuttal. In particular, the reviewer mentions the lack of support for the narrative of the paper, since many conclusions remain vague (e.g., "Our method offers insights into the spectrum distributions [...]"), and in particular it is not clear what a practitioner can learn in terms of GCNs (since, for example, the real-world experiments do not agree with the toy experiments). I generally agree with all the concerns expressed by the reviewer.

In addition, the paper has language, figures, and notation issues, including weird choices of adjectives ("impeccable performance", "prowess of MP", "exacerbating performances") and disconnected sections.

**Resubmission Of Major Revision:**

The authors may consider submitting a major revision at a later time.